# Learning Permutation-invariant Macroscopic Dynamics

**Zhichao Han**[1]   **Mengyi Chen**[2]   **Qianxiao Li**[2 1]

## Abstract

Accurately modeling the macroscopic dynamics of high-dimensional microscopic systems is of broad interest across the sciences. Many data-driven approaches learn a low-dimensional latent state through an autoencoder trained for pointwise input reconstruction. These methods typically assume a fixed ordering of microscopic degrees of freedom in the input. However, in many settings, such as particle systems, the microscopic state is inherently unordered. This motivates an autoencoder framework that learns permutation-invariant latent representations. To this end, we adopt a permutation-invariant encoder and design the decoder to reconstruct the mass distribution centered at the observed points rather than per-sample reconstruction. We then jointly learn the macroscopic dynamics of the observables together with the latent states. We demonstrate the effectiveness and robustness of the proposed method across a range of microscopic settings, including learning the energy dynamics in interacting particle systems, predicting mixing dynamics in Lennard–Jones fluids, and modeling the stretching dynamics from video data of polymers moving in an elongational force field.

## 1. Introduction

Given high-dimensional microscopic observations of a physical system, accurately modeling its macroscopic dynamics is of broad interest across disciplines (Givon et al., 2004). A central challenge is to construct a closed macroscopic model: the evolution of macroscopic quantities depends on microscopic degrees of freedom, and we therefore seek closure variables that encode the microscopic information.

[1]Institute for Functional Intelligent Materials, National University of Singapore [2]Department of Mathematics, National University of Singapore. Correspondence to: Qianxiao Li <qianxiao@nus.edu.sg>.

*Proceedings of the $43^{rd}$ International Conference on Machine Learning*, Seoul, South Korea. PMLR 306, 2026. Copyright 2026 by the author(s).

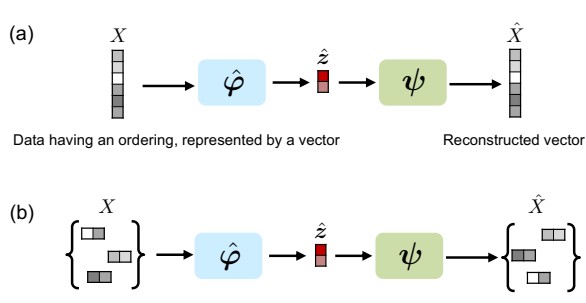

*Figure 1.* Existing and desired autoencoder for learning closure variables. (a) Standard autoencoder to learn the latent $\hat{z}$. Microstate $X$ is represented by a vector based on the ordering. The encoder $\hat{\varphi}$ and decoder $\psi$ are typically implemented as MLPs. (b) The desired autoencoder to learn $\hat{z}$. The model should preduce a permutation-invariant latent variable $\hat{z}$ for inputs that lack the canonical ordering.

Recent data-driven approaches leverage neural networks to automatically discover low-dimensional descriptions that serve as closure variables. These methods have achieved notable success for many scientific applications, including learning reduced thermodynamic coordinates and predicting stretching length dynamics of polymers (Chen et al., 2023b), discovering slow collective variables and reduced kinetic models for molecular conformational dynamics (Wehmeyer & Noé, 2018; Mardt et al., 2018), and learning macroscopic internal variables for history-dependent heterogeneous materials (Liu et al., 2023).

Most existing approaches learn closure variables using an autoencoder trained to minimize the pointwise reconstruction loss. The encoder and decoder are typically parameterized by the Multilayer Perceptron (MLP) (Chen et al., 2023b; Chen & Li, 2024) or Convolutional Neural Network (CNN) (Lee & Carlberg, 2020a), assuming that the microscopic degrees of freedom admit a fixed ordering to be represented as vectors or tensors (Fig. 1(a)). However, many physical systems have no intrinsic ordering, such as a set of interacting particles. The desired closure variable must be *permutation-invariant* with respect to the input (Fig. 1(b)). Applying existing methods, which require a canonical ordering on the input, can be prohibitively difficult in practice.

This gap arises because the autoencoders parameterized by

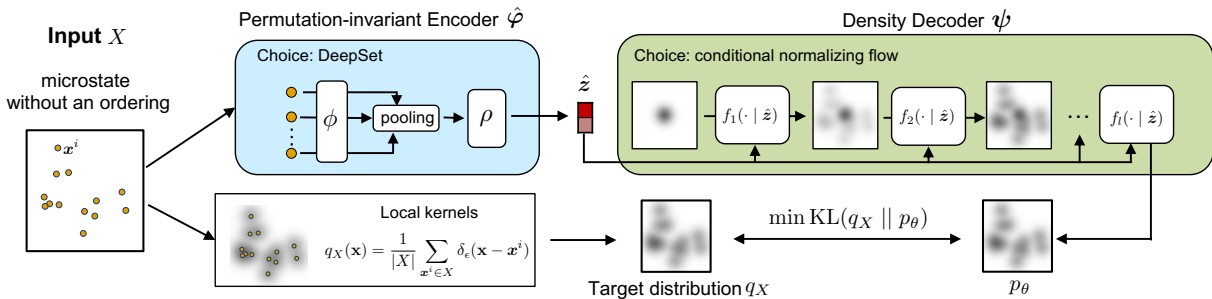

*Figure 2.* Overview of our distribution-aware autoencoder for closure modeling. The encoder $\hat{\varphi}$ maps unordered microstate $X$ to a permutation-invariant latent variable $\hat{z}$. The microstate $X$ induces a target distribution $q_X$. Conditioned on $\hat{z}$, the decoder $\psi$ generates a density $p_\theta(\mathbf{x}|\hat{z})$ to approximate the target density $q_X$. In general, $\hat{\varphi}$ is a permutation-invariant set function and $\psi$ is a conditional density function. We instantiate $\hat{\varphi}$ with DeepSet and $\psi$ with the conditional normalizing flow.

MLP or CNN are not symmetric to permutation. The input and output are both indexed by an ordering, and the usual pointwise reconstruction loss fixes an index-by-index correspondence. Consequently, the learned latent vector encodes the input information with a specific ordering, and can change when the same microstate is presented under a different permutation. One may consider adopting models for sets (e.g., DeepSet (Zaheer et al., 2017) or Set Transformer (Lee et al., 2019)) as the encoder to produce a permutation-invariant latent variable. Yet, a fundamental difficulty remains on the decoding side: without a specified ordering, there is no natural mechanism to reconstruct the microstate from a single low-dimensional embedding using an index-wise loss. For instance, consider a microstate $\{\boldsymbol{x}^i\}_{i=1}^N$ with $N = 3$ particles. If a decoder is trained with a pointwise MSE loss to output an ordered list $(\hat{\boldsymbol{x}}^1, \ldots, \hat{\boldsymbol{x}}^N)$, the loss will penalize $(\hat{\boldsymbol{x}}^1, \hat{\boldsymbol{x}}^2, \hat{\boldsymbol{x}}^3)$ and $(\hat{\boldsymbol{x}}^3, \hat{\boldsymbol{x}}^2, \hat{\boldsymbol{x}}^1)$ differently, even though they correspond to the same physical configuration. Crucially, the number of such physically equivalent permutations scales as $N!$. Therefore, training often relies on explicit matching (Zhang et al., 2019; Rezatofighi et al., 2021) or permutation-invariant losses (Achlioptas et al., 2018; Yang et al., 2018); both add substantial computational cost (Nguyen et al., 2021) and may introduce optimization instability (Li et al., 2022; Sharifipour et al., 2025).

To overcome this difficulty, we propose an alternative reconstruction objective for closure modeling. Rather than reconstructing the microscopic degrees of freedom directly, we reconstruct their distribution. Concretely, we induce a mass distribution concentrated at the observed points, and train the decoder to reconstruct this density from the latent representation. This distributional reconstruction eliminates the need to align individual points or impose an ordering, thereby enabling learning of permutation-invariant latent variables. The learned latent variable summarizes the input distribution and serves as closure variables for modeling macroscopic dynamics. Our contributions can be summarized as follows.

- We propose a novel strategy for closure modeling that trains an autoencoder to reconstruct the *distributional information* instead of pointwise reconstruction.

- We demonstrate the effectiveness and robustness of the proposed approach across multiple microscopic settings, including learning the energy dynamics of interacting particle systems, the mixing dynamics of Lennard–Jones fluid and predicting the stretching dynamics of polymers from videos.

## 2. Related Work

**Learning Reduced Dynamics** Reduced-order models (ROMs) and closure modeling approximate high-dimensional dynamical systems using low-dimensional latent variables and learn latent dynamics, often with the goal of improving computational efficiency (Schilders et al., 2008; Benner et al., 2015). ROMs typically aim to approximate the full-order state by evolving latent variables and decoding them back to the full-order space (Carlberg et al., 2017; Lee & Carlberg, 2020b; Chen et al., 2023a). In contrast, closure modeling targets accurately modeling the dynamics of selected macroscopic observables and, in principle, does not have to reconstruct the full-order state (Sagaut, 2006; Chen et al., 2023b; Boral et al., 2023). Nevertheless, many data-driven closure approaches adopt a reconstruction objective to learn closure coordinates. The reconstruction loss offers a convenient objective that enforces the latent variables to encode microstate information and regularizes the latent variable (Champion et al., 2019; Chen et al., 2023b; Wan et al., 2023; Zhu et al., 2025). Accordingly, existing closure modeling methods often consist of two components: (i) an autoencoder for discovery of closure variables trained with a reconstruction loss and (ii) a reduced dynamical model for predicting the latent dynamics (Champion et al., 2019; Chen et al., 2023b; Wan et al., 2023; Zhu et al., 2025). Our method belongs to closure modeling, as we focus on modeling the dynamics of macroscopic quantities and leverage a reconstruction-based objective to learn clo-

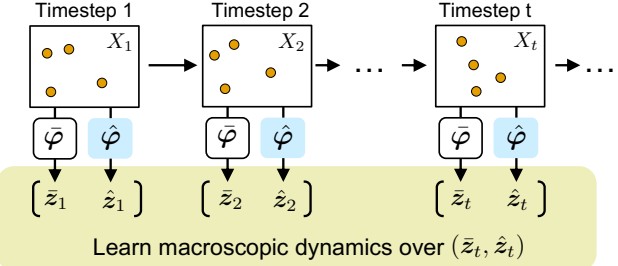

*Figure 3.* Workflow of learning macroscopic dynamics. We are given high-dimensional observations at the microscopic scale that evolve over time. The deterministic encoder $\bar{\varphi}$ extracts the macroscopic feature of interest $\bar{z}$ (for example, the system energy). A learned encoder $\hat{\varphi}$ extracts the closure macroscopic variables $\hat{z}$. A dynamic model is trained to predict the macroscopic dynamics of $\bar{z}$ and $\hat{z}$ together.

sure variables. Prior work has mainly focused on dynamical systems whose state variables have a fixed indexing, such as discretized partial differential equation (PDE) solutions on grids (Champion et al., 2019; Pant et al., 2021; Chen et al., 2023a). However, these methods are not directly applicable when the microstate has no canonical ordering. Our method addresses this challenge through a permutation-invariant architecture and a newly designed permutation-invariant training objective.

**Learning Permutation-invariant Representations for Sets** General machine learning models have been proposed to learn permutation-invariant representations of sets. Classic examples include pooling-based architectures such as DeepSet (Zaheer et al., 2017) and PointNet (Qi et al., 2017), as well as attention-based models such as the Set Transformer (Lee et al., 2019). In our model, we adopt their architecture as the encoder. Unlike many previous works that use set encoders as standalone modules for supervised prediction (Zaheer et al., 2017; Lee et al., 2019; Ilse et al., 2018; Qi et al., 2017; Skianis et al., 2020), we instead learn permutation-invariant closure variables by pairing the set encoder with a decoder and optimizing an unsupervised reconstruction objective. Designing the proper decoder is non-trivial because the input has no ordering. A naive decoder that outputs individual elements raises the question of how to align reconstruction with the input. To do this, one has to explicitly handle element ordering, commonly via permutation-invariant set-matching losses (as commonly done in point-cloud reconstruction; see next paragraph). We avoid this complication by reconstructing the *input distribution*, thereby removing any dependence on the ordering.

**Point Clouds Reconstruction** Various autoencoder architectures have been proposed for point cloud reconstruction (Achlioptas et al., 2018; Deng, 2018; Groueix et al., 2018; Yang et al., 2018; 2021; Yu et al., 2018; Zhao et al., 2019). These methods utilize permutation-invariant architectures as the encoder (as described above) and design

the decoder that outputs an explicit set of points intended to match the input samples. To make the reconstruction invariant to how the points are indexed, they either introduce a combinatorial matching step (Zhang et al., 2019; Rezatofighi et al., 2021) or optimize permutation-invariant distances (e.g., Chamfer discrepancy, Earth Mover's distance, and sliced Wasserstein distance) as reconstruction loss (Achlioptas et al., 2018; Yang et al., 2018). More recently, Kilgour et al. (2025) proposed a soft-matching reconstruction objective by representing both the input and output as point-centered Gaussian mixtures and measuring reconstruction accuracy via pairwise Gaussian overlaps. While our approach also learn permutation-invariant latent variables from a reconstruction objective, it differs in *what* is reconstructed. Rather than decoding an explicit point set and performing pointwise (hard or soft) alignment, we decode a *density* that describes the microstate. Concretely, we induce a mass distribution concentrated at the observed points and train the decoder to reconstruct this density from the latent variable. This formulation is inherently permutation-invariant and avoids the matching issue for unordered points.

## 3. Method

### 3.1. Problem Statement

Let $X_t = \{\boldsymbol{x}_t^1, \boldsymbol{x}_t^2, \ldots, \boldsymbol{x}_t^n\} \in \mathcal{X}$ be the state of $n$ *unordered* particles at time $t$, where $\boldsymbol{x}_t^i \in \mathbb{R}^d$ is the state of particle with index $i$. In many real-world systems, the number of particles $n$ is very large, and the microscopic system is therefore high-dimensional. A deterministic function $\bar{\varphi}$ is given beforehand to extract the macroscopic quantities of interest from the microscopic state, $\bar{\varphi} : \mathcal{X} \to \bar{\mathcal{Z}}$ where $\bar{\mathcal{Z}}$ is a space containing low-dimensional vectors. We restrict our discussion to intensive macroscopic observables such as the average energy of the interacting particle system, so that their magnitude does not scale with the system size $n$. The goal of our method is to learn a permutation-invariant macroscopic dynamical model that can be applied to microscopic systems with varying numbers of particles.

To this end, an autoencoder is used to learn a low-dimensional representation that serves as the closure variable, together with a latent dynamical model for predicting macroscopic dynamics. The architecture is illustrated in Fig. 3. The encoder, $\hat{\varphi} : \mathcal{X} \to \hat{\mathcal{Z}}$, maps the microstate $X \in \mathcal{X}$ to a low-dimensional latent vector $\hat{z} = \hat{\varphi}(X)$. The latent $\hat{z}$ should be invariant to the ordering of $X = \{\boldsymbol{x}^1, \boldsymbol{x}^2, \ldots, \boldsymbol{x}^n\}$, i.e.,

$$\hat{\varphi}(\sigma X) = \hat{\varphi}(X), \quad \forall \sigma \in S_n, \tag{1}$$

where $S_n$ denotes the symmetric group on $n$ elements and $\sigma X := \{\boldsymbol{x}^{\sigma(1)}, \boldsymbol{x}^{\sigma(2)}, \ldots, \boldsymbol{x}^{\sigma(n)}\}$ is the permuted particle set. In parallel, we extract the macroscopic quantity of interest $\bar{z}_t = \bar{\varphi}(X_t)$, and form the closed macroscopic state

$\boldsymbol{z}_t = [\bar{\boldsymbol{z}}_t, \hat{\boldsymbol{z}}_t] \in \mathcal{Z}$. We denote the dimension of $\hat{\boldsymbol{z}} \in \hat{\mathcal{Z}}$, $\boldsymbol{z} \in \mathcal{Z}$ by $\hat{z}_{\text{dim}}$, $z_{\text{dim}}$. Next, we describe the closure modeling to learn $\hat{\boldsymbol{z}}$ and the macroscopic dynamics modeling that predicts the evolution of $\boldsymbol{z}_t$.

## 3.2. Closure Modeling

As discussed in Sec. 1, a common approach is to train an autoencoder using a *per-sample* reconstruction loss and to use the learned latent representation as the closure variable. That is, they represent $\{\boldsymbol{x}_t^1, \boldsymbol{x}_t^2, \ldots, \boldsymbol{x}_t^n\}$ by a vector $\boldsymbol{y} \in \mathbb{R}^{nd}$ according to an indexing and minimize an element-wise reconstruction error of the form:

$$\mathbb{E}_{\boldsymbol{y}}\left[\sum_{i=1,2,\ldots,nd} l\Big(\boldsymbol{y}[i], \boldsymbol{\psi}(\hat{\boldsymbol{\varphi}}(\boldsymbol{y}))[i]\Big)\right] ,$$

where $[i]$ denotes the i-th entry in a vector and $l$ is the loss such as squared error. However, when the input has no canonical ordering, the index-wise loss above is not permutation-invariant and can penalize reconstructions that are correct up to permutation. We resolve this by shifting the reconstruction objective from input particles themselves to a *distributional* representation of the input set.

In our implementation (Fig. 2), we parameterize the encoder $\hat{\boldsymbol{\varphi}}$ using DeepSet to obtain a permutation-invariant representation $\hat{\boldsymbol{z}}$ (Eq. 1). While we use DeepSet in this work, other permutation-invariant set encoders such as set transformer (Lee et al., 2019) can be used as well. For each $X \in \mathcal{X}$, we induce a continuous density $q_X(\mathbf{x})$ over the input space $\mathcal{X}$ that corresponds to a mass distribution centered at the observation points, i.e.,

$$q_X(\mathbf{x}) = \frac{1}{|X|} \sum_{\boldsymbol{x}^i \in X} \delta_\epsilon(\mathbf{x} - \boldsymbol{x}^i), \tag{2}$$

where $\delta_\epsilon(\cdot)$ is an isotropic Gaussian kernel with variance $\epsilon^2$. The $q_X(\mathbf{x})$ serves as the reconstruction target. The parameter $\epsilon$ acts as a bandwidth (smoothing scale) and therefore controls the resolution at which the set is represented: a smaller $\epsilon$ yields sharper, more localized modes around individual particles, while a larger $\epsilon$ produces a smoother density that discards more local details. More discussions on $\epsilon$ are in Appendix B.3 and the numerical results are in Appendix C.2.

Given the learned $\hat{\boldsymbol{z}} = \hat{\boldsymbol{\varphi}}(X)$ and the target density $q_X(\mathbf{x})$, the decoder $\psi$ generates a conditional density $p(\mathbf{x}|\hat{\boldsymbol{z}})$ via

$$p_\theta(\mathbf{x}|\hat{\boldsymbol{z}}) = \boldsymbol{\psi}(\mathbf{x}; \hat{\boldsymbol{\varphi}}(X)) ,$$

where $\theta$ denotes all learnable parameters in $\hat{\boldsymbol{\varphi}}$ and $\psi$. We implement $p(\mathbf{x}|\hat{\boldsymbol{z}})$ with a conditional normalizing flow, though any conditional density model can be used (Chen et al., 2018; Lipman et al., 2023). We refer readers to Appendix B.2 for further implementation details. In this work,

we use the KL divergence to measure the discrepancy between $q_X(\mathbf{x})$ and $p_\theta(\mathbf{x}|\hat{\boldsymbol{z}})$, although many alternative distributional losses could be used (Arjovsky et al., 2017):

$$\mathcal{L}_{\text{rec}} = \mathbb{E}_X \left[\text{KL}(q_X(\mathbf{x}) \parallel p_\theta(\mathbf{x}|\hat{\boldsymbol{z}}))\right] . \tag{3}$$

We estimate the KL divergence with Monte Carlo samples drawn from $q_X(\cdot)$. Noting that $q_X(\cdot)$ is a mixture of Gaussians sharing the same variance and weight, we can efficiently draw samples in parallel by firstly sampling the mixture component uniformly and then sampling from the local Gaussian.

This designed autoencoder brings several advantages for closure modeling. First, the learned $\hat{\boldsymbol{z}}$ is permutation-invariant to input indexing. Second, the encoder admits variable-size inputs, enabling $\hat{\boldsymbol{z}}$ to be computed for different particle counts. Third, the training objective avoids explicit set-to-set matching, which could be computationally expensive. Specifically, in our pipeline, the DeepSet encoder scales linearly with the input size, i.e., $\mathcal{O}(n)$ with respect to the number of input particles $n$, because the encoder processes each particle feature independently before aggregation. Given the latent variable generated by the encoder, the computational complexity of the density-based decoder is $\mathcal{O}(1)$ with respect to the input size. This is because the reconstruction objective is evaluated using Monte Carlo samples from the target density, and the sampling procedure is independent of the number of particles under our construction. Therefore, unlike pointwise reconstruction losses that may require expensive pointwise matching, the computational cost of the proposed distribution-reconstruction objective depends on the number of Monte Carlo samples, the input feature dimension, and the cost of normalizing flow evaluations, rather than the number of input particles.

## 3.3. Macroscopic Dynamics Modeling

We model the dynamics of the augmented macroscopic state $\boldsymbol{z}_t = [\bar{\boldsymbol{z}}_t, \hat{\boldsymbol{z}}_t]$ by

$$d\boldsymbol{z}_t = \boldsymbol{g}(\boldsymbol{z}_t)\,dt + \boldsymbol{\Sigma}(\boldsymbol{z}_t)\,d\boldsymbol{W}_t, \tag{4}$$

where $\boldsymbol{g}(\boldsymbol{z}_t)$ is the drift term, $\boldsymbol{\Sigma}(\boldsymbol{z}_t)$ is the diffusion term, and $\boldsymbol{W}_t$ denotes standard Brownian motion. Since the input consists of discrete frames, we learn a parameterized dynamics model by maximizing the likelihood of one-step transitions implied by an Euler–Maruyama discretization with step size $\Delta t$:

$$p_{\boldsymbol{g},\boldsymbol{\Sigma}}(\boldsymbol{z}_{t+1}|\boldsymbol{z}_t) = \mathcal{N}\Big(\boldsymbol{z}_t + \boldsymbol{g}(\boldsymbol{z}_t)\,\Delta t, \;\; \Delta t\,\boldsymbol{\Sigma}(\boldsymbol{z}_t)\boldsymbol{\Sigma}(\boldsymbol{z}_t)^\top\Big) ,$$

Where $p_{\boldsymbol{g},\boldsymbol{\Sigma}}$ indicates the likelihood depends on the parameters in $\boldsymbol{g}$ and $\boldsymbol{\Sigma}$. We therefore train $(\boldsymbol{g}, \boldsymbol{\Sigma})$ by minimizing the negative conditional log-likelihood

$$\mathcal{L}_{\text{dyn}} = \mathbb{E}_{\boldsymbol{z}_t, \boldsymbol{z}_{t+1}}[-\log p_{\boldsymbol{g},\boldsymbol{\Sigma}}(\boldsymbol{z}_{t+1}|\boldsymbol{z}_t)] . \tag{5}$$

In the deterministic case, we set $\Sigma(z) \equiv 0$ and minimizes a standard one-step squared-error loss.

Learning the dynamics over $z$ naturally yields the macroscopic dynamics of $\bar{z}$, which is our quantity of interest. In this work, we parameterize the drift $g$ and diffusion $\Sigma$ if it is present with MLPs, but any dynamical model can be used as well, such as the OnsagerNet (Yu et al., 2021). Once trained, the encoder is used only to construct the initial state $z_0$. Then the learned dynamics model generates the trajectory *autoregressively*, i.e., it predicts future macroscopic states $z_1, z_2, \ldots, z_t$ given only an initial state $z_0$.

**Overall Objective** We train the full framework with the total loss

$$\mathcal{L} = \mathcal{L}_{\text{rec}} + \lambda_{\text{dyn}}\mathcal{L}_{\text{dyn}},$$

where $\lambda_{\text{dyn}}$ balances distribution reconstruction and dynamical consistency. Following existing approaches (Champion et al., 2019; Greydanus et al., 2019), we use a two-stage training procedure to learn the latent states and their corresponding latent dynamical models separately. The pseudocode of training and inference procedures is provided in Appendix A.1.

## 4. Experiments

We compare our method against prior closure modeling approaches to assess its effectiveness. As noted, most existing works assume ordered microstates and employ an autoencoder with MLPs for the encoder and decoder, trained by minimizing a pointwise reconstruction loss (Champion et al., 2019; Fries et al., 2022; Chen & Li, 2024). In contrast, we consider permutation-invariant closure modeling from unordered microstates. To enable a fair comparison, we adopt the common strategy of sampling random input permutations during training (Li et al., 2018). During training, we randomly permute the order of the input anchors within each mini-batch before feeding them to the flattened MLP autoencoder. Combining this permutation-augmentation strategy with a standard MLP autoencoder trained using pointwise MSE reconstruction loss gives our first baseline, denoted as **(AE-Aug)**. Besides, we also replace the MLP encoder with a symmetric architecture to learn permutation-invariant closure variables. Concretely, we use a DeepSet encoder for fair comparison paired with an MLP decoder, and train the autoencoder using a pointwise MSE reconstruction loss. We denote this baseline as **(AE-InvE)**. Moreover, since point clouds reconstruction (see Sec. 2) can also generate permutation-invariant latent variables, we include an additional baseline aligned with that literature for a more comprehensive comparison. Concretely, we use the same DeepSet encoder and MLP decoder but train the autoencoder with the Chamfer distance as the reconstruction loss (Achlioptas et al., 2018). We denote this baseline as **(AE-InvE-CD)**. Finally, because closure modeling

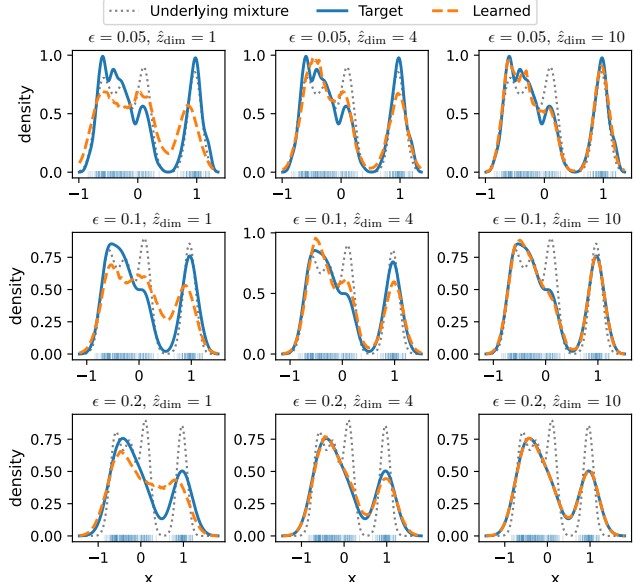

*Figure 4.* Distributional reconstruction with different $\epsilon$ and $\hat{z}_{\text{dim}}$. The gray dotted curve denotes the Gaussian mixture distribution from which particles are sampled, and the blue rug marks indicate sample locations. The blue solid curve shows the target density. The orange dashed curve is the density learned by our model.

does not necessarily require a reconstruction objective, we also consider the strategy of directly training an encoder jointly with a latent dynamics model. In practice, however, this optimization can be ill-posed: the latent representation may collapse, with the encoder producing constant or otherwise non-informative vectors (Tang et al., 2023; Saanum et al., 2024). We nevertheless implement this approach as a baseline **(InvE)**, where a DeepSet encoder generates the permutation-invariant latent variables, which are then concatenated with the macroscopic features of interest and passed to the dynamical model.

We evaluate our method and baselines on macroscopic variable prediction. Concretely, given an initial macroscopic state $z_0$ generated by $\bar{\varphi}$ and the trained $\hat{\varphi}$ on $X_0$, we apply the trained latent dynamics model (Eq. 4) to recursively predict future states $z_1, z_2, ..., z_t$. We then compare the predicted and the ground-truth $\bar{z}_1, \bar{z}_2, ..., \bar{z}_t$. For deterministic dynamics, we report the mean relative error (MRE) (Fries et al., 2022); for stochastic dynamics, we use the maximum mean discrepancy (MMD) (Gretton et al., 2012; Kidger et al., 2021). Details are provided in Appendix A.2. We ensure that all models have the same latent-variable dimension for $\hat{z}$ per experiment to enable a fair comparison.

### 4.1. Visualizing Distributional Reconstruction

We first analyze the reconstruction of our distribution-aware autoencoder as we vary $\epsilon$ in the target distribution $q_X(\mathbf{x})$

and the encoder dimension $\hat{z}_{\dim}$. To this end, we construct a 1D synthetic dataset comprising multiple particle sets. For each choice of $\epsilon$, we induce the target distribution and minimize Eq. 3 to learn $\hat{z}$ under different values of $\hat{z}_{\dim}$ Specifically, we generate 3300 sets, using 3000 for training and 300 for validation. Each set consists of 200 particles sampled from a Gaussian mixture distribution, with mixture means and variances varying across sets. For each $\epsilon$, we train our model with different $\hat{z}_{\dim}$ to approximate the induced target distribution, and the results are reported in Fig. 4.

Results indicate that the encoder bottleneck $\hat{z}_{\dim}$ strongly affects reconstruction quality. When $\hat{z}_{\dim}$ is small (e.g., $\hat{z}_{\dim} = 1$ in this case), the model frequently underfits the target: the learned density is overly smooth and fails to match the multimodal structure. Increasing $\hat{z}_{\dim}$ improves reconstruction, enabling the learned distribution closely reconstruct the target density. We also observe a clear effect that a smaller $\epsilon$ yields a more local and sharper target density (top row). Larger $\hat{z}_{\dim}$ is required to reconstruct these details. As $\epsilon$ grows (middle and bottom rows), the target becomes smoother, and correspondingly, the reconstruction becomes easier. A moderate latent dimension $\hat{z}_{\dim}$ can achieve near-target fits. Overall, the figure demonstrates how $\epsilon$ controls the smoothness of the target distribution, and a small $\hat{z}_{\dim}$ is enough for a smooth target density.

### 4.2. Energy Evolution of Interacting Particle Systems

We study the energy evolution in particle systems. For this, we adopt the 2D interacting particle system from Kolokolnikov et al. (2011), where each particle interacts with all others through a pairwise step-force law $F(r) = \tanh(a(1 - r)) + b$. We set $a = 4$ and $b = 0.1$, and simulate trajectories using a forward Euler integrator with time step $\Delta t = 0.002$. The initial particle positions are sampled from a two-component Gaussian mixture model, with component means and isotropic covariances randomly drawn per trajectory. The 2D particle positions serve as microstates $X$, and the $\bar{\varphi}$ computes the *normalized pairwise interaction energy*, which is the total pairwise interaction energy divided by the number of particle pairs (see Appendix A.3 for the energy computation). Fig. 5(a) visualizes a examplary trajectory: particles evolve from the initial configuration to a spatial pattern. Correspondingly, the normalized interaction energy decreases over time and converges to a steady value.

We generate 10k simulated trajectories and split them into training/validation sets with an $8/2$ split. Each trajectory contains 300 particles recorded at 301 time steps. Since this macroscopic evolution is deterministic, we model $z_t$ using an ODE (i.e., setting $\Sigma(z) = 0$ in Eq. 4). We evaluate macroscopic prediction in three test regimes: First, we generate 1k trajectories with the same number of particles as training, and sample initial positions also from a two-

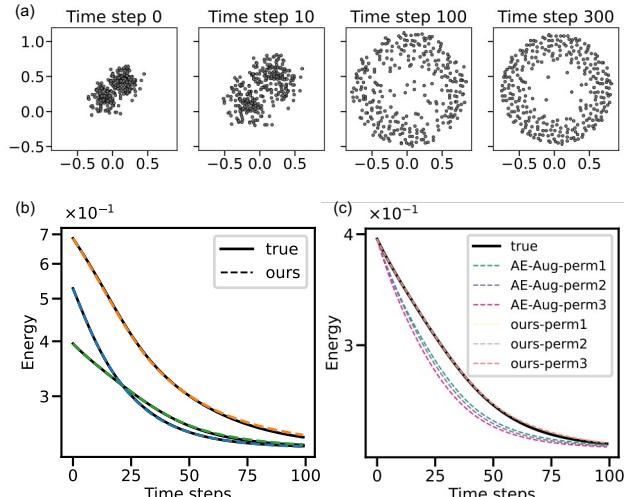

*Figure 5.* Experiment on the interacting particle system. (a) Microstate snapshots examples. (b) $\bar{z}$ predicted by our model versus the ground truth. (c) Permutation-invariant verification.

component GMM with independently resampled component means and isotropic covariances. This is for in-distribution evaluation, and we denote it as *in-dst*. Secondly, we generate 1k trajectories with the same number of particles as training, but sample initial positions from a three-component GMM. This evaluates whether the learned macroscopic dynamics is generalizable to different initial patterns, and we denote it as *diff-init*. Third, we generate 1k trajectories initialized from the same two-component GMM pattern but with more particles (400), denoted as *diff-N*. It evaluates whether the learned models are generalizable to different input sizes.

*Table 1.* Mean relative error on predicting energy evolution in the interacting particle system. The mean and standard deviation are computed from three runs.

| | in-dst | diff-init | diff-N |
|---|---|---|---|
| AE-Aug | $1.25 _{\pm 0.016} \times 10^{-3}$ | $8.04 _{\pm 0.10} \times 10^{-4}$ | N/A |
| AE-InvE | $2.41 _{\pm 0.34} \times 10^{-4}$ | $5.87 _{\pm 0.49} \times 10^{-4}$ | $2.49 _{\pm 0.43} \times 10^{-4}$ |
| AE-InvE-CD | $6.14 _{\pm 0.73} \times 10^{-5}$ | $\mathbf{3.90} _{\pm 0.12} \times 10^{-4}$ | $6.43 _{\pm 1.10} \times 10^{-5}$ |
| InvE | $6.01 _{\pm 0.49} \times 10^{-5}$ | $4.85 _{\pm 0.11} \times 10^{-4}$ | $6.13 _{\pm 0.30} \times 10^{-5}$ |
| ours | $\mathbf{5.19} _{\pm 0.25} \times 10^{-5}$ | $4.38 _{\pm 0.19} \times 10^{-4}$ | $\mathbf{5.22} _{\pm 0.46} \times 10^{-5}$ |

We first test the macroscopic variable prediction by our model in Fig. 5(b). Specifically, we select three trajectories from the *in-dst* test set, provide only their initial macroscopic state $z_0$, and recursively predict the future states. The prediction closely matches the ground truth throughout the rollout, capturing both the rapid initial decay and the convergence to a steady value. Furthermore, we test whether the learned macroscopic variable is permutation invariant. We take one *in-dst* test trajectory, apply three random permutations of the particle ordering, and compare the macroscopic prediction from our model and AE-Aug in Fig. 5(c). AE-Aug produces noticeably different energy predictions

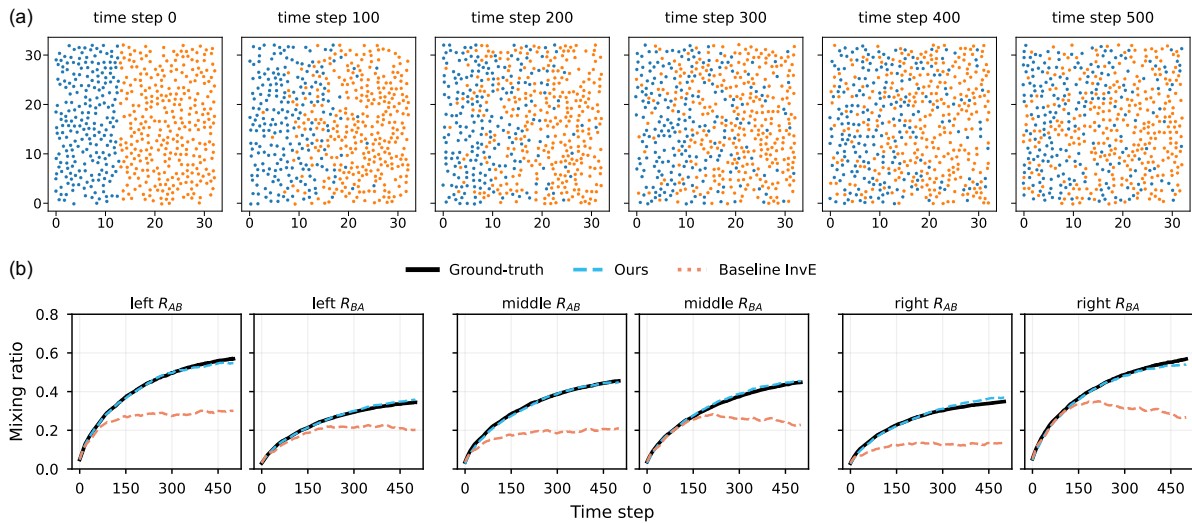

*Figure 6.* Experiment on the binary particle mixing. (a) An example microstate trajectory for illustrating the mixing process. (b) The mean of $R_{AB}$ and $R_{BA}$ under three initial separation boundaries: $x = 12$ (left), $x = 16$ (middle), and $x = 20$ (right).

under different permutations, even with permutation augmentation during training. In contrast, our method is exactly permutation-invariant: the three prediction curves overlap, appearing as a single curve. We then compare our method to baselines quantitatively in Table 1. On *in-dst*, our method achieves the lowest error, substantially improving over the plain AE-Aug baseline. On the particle-count shift (*diff-N*), our method again attains the best performance, suggesting that the learned macroscopic dynamics transfers across different system sizes. Under the initialization-pattern shift test (*diff-init*), our method remains competitive and improves over AE-Aug and AE-InvE, while AE-InvE-CD attains the lowest error. We also examine the influence of $\hat{z}$ dimension and the smoothing scale $\epsilon$, and the sensitivity test is provided in Appendix C.2.

### 4.3. Mixing of Binary Particles

We next study a stochastic multi-species system where the macroscopic state quantifies the *mixing* of two types of particles. Specifically, we consider a two-dimensional binary particle system mixing under a pairwise Lennard-Jones potential, a standard model in atomistic binary fluids (Das et al., 2003; Yaneva et al., 2005). We simulate 529 particles in a square domain of side length $l = 32$ with reflective boundaries in both $x$ and $y$. At initialization, particles are randomly placed, and a vertical boundary is placed at a random location to partition the domain into two regions: particles on one side are labeled type A and those on the other side are labeled type B (Fig. 6(a) left). We run the NVT simulation at temperature $1.0$ for $10^5$ steps and record particle positions every 200 steps as microstates $X_t$. Fig. 6 (a) shows a representative trajectory, where the initially separated particles gradually interpenetrate until reaching a

well-mixed configuration. We compute the mixing ratio as the macroscopic variable $\bar{z}$, i.e., $\bar{z} = (R_{AB}, R_{BA})$, where $R_{AB}$ is the probability that a neighbor of a type-A particle is type-B (and vice versa for $R_{BA}$), computed with a fixed cutoff radius. The mixing ratio is equivalent to the short-range order measure commonly used in chemistry (Sheriff et al., 2024). Intuitively, $\bar{z}$ increases as mixing progresses and converges when the two species become uniformly mixed. We generate 10k trajectories with sampling the initial separation boundary uniformly in $[10, 22]$, and use the $8/2$ split for training and validation. Similar to the previous experiment, we evaluate the rollout prediction in three regimes. First, we generate 200 test trajectories with the separation boundary sampled in the same range $[10, 22]$, denoted as *in-dst*. Then, we generate test data *diff-N* with a reduced domain height while keeping density fixed, resulting in 437 particles per simulation. The separation boundary is also sampled in $[10, 22]$. Finally, we simulate 200 trajectories with the separation boundary sampled outside the training range (uniformly in $[8, 10] \cup [22, 24]$), denoted as *diff-init*.

*Table 2.* The MMD on predicting the mixing ratio. Mean and standard deviation are computed across three runs.

|  | in-dst | diff-init | diff-N |
|---|---|---|---|
| AE-Aug | $1.91_{\pm 0.22} \times 10^{-2}$ | $1.07_{\pm 0.045} \times 10^{-1}$ | N/A |
| AE-InvE | $2.60_{\pm 0.46} \times 10^{-2}$ | $9.17_{\pm 0.47} \times 10^{-2}$ | $5.26_{\pm 0.75} \times 10^{-2}$ |
| AE-InvE-CD | $2.24_{\pm 0.92} \times 10^{-2}$ | $1.04_{\pm 0.050} \times 10^{-1}$ | $2.16_{\pm 0.99} \times 10^{-2}$ |
| InvE | $1.43_{\pm 0.049} \times 10^{-1}$ | $1.22_{\pm 0.048} \times 10^{-1}$ | $1.41_{\pm 0.053} \times 10^{-1}$ |
| ours | $\mathbf{1.09}_{\pm 0.11} \times 10^{-2}$ | $\mathbf{5.70}_{\pm 0.14} \times 10^{-2}$ | $\mathbf{9.64}_{\pm 0.084} \times 10^{-3}$ |

To accommodate two particle types, our method applies a *shared* encoder and a *shared* decoder to each species separately, and then aggregates these two embeddings to form the closure variable $\hat{z}$ for the full system. For a fair

comparison, we configure each baseline to use particle-type information whenever its architecture permits; otherwise, we train it on the full particle set without distinguishing types (see Appendix A.4 for details). Because of the thermal fluctuation, we model the dynamics of $z_t$ with an SDE (i.e., learning both $g$ and $\Sigma$ in Eq. 4).

The quantitative comparisons between our methods and baselines are reported in Table 2. Our method achieves the lowest MMD on all test settings. Comparing *diff-init* to *in-dst*, MMD increases for all methods because the macrostates are unseen during training, but our method still performs best. The MMD is lower for *diff-N* than for *in-dst* for all methods because the system of *diff-N* is a subset of the system used in training. Moreover, we evaluate macroscopic predictions in three representative settings by placing the initial separation boundary on the left, middle, or right of the domain at $x \in 12, 16, 20$. For each setting, we generate 100 trajectories with randomly sampled initial particle positions. Starting from the initial macrostates $z_0$, we compare the predicted $z_t$ to the ground-truth. We compare our method to baseline InvE, which is the reconstruction-free method. The predictions are as shown in Fig. 6 (b). Across all three cases, the means of our predictions are close to the ground-truth means for both $R_{AB}$ and $R_{BA}$, whereas InvE underestimates the mixing ratios and tends to plateau prematurely. These results support the observation that closure modeling does not necessarily require reconstruction in principle, but directly learning latent dynamics can be challenging because the representation may collapse (Tang et al., 2023; Saanum et al., 2024). Careful initialization of learnable parameters or additional stabilization techniques may alleviate this issue, and we leave it for future work.

### 4.4. Polymer Extension

Besides particle systems, many scientific applications provide data in image format (Chen et al., 2022; Botev et al., 2021). We therefore test whether our method can learn closure variables for macroscopic dynamics modeling from videos of microstate trajectories, even though it is not specifically designed for image inputs. To this end, we consider the polymer dynamics dataset from (Chen et al., 2023b), which simulates the dynamics of polymer chains using the Brownian dynamics approach in a planar elongational flow. We convert each polymer frame (3D bead coordinates) into a 2D image by rendering a Gaussian blob at each bead's (x, y) coordinate, with the blob width determined by the z coordinate. Additional details on the original dataset and our image construction procedure are provided in Appendix A.5.

To apply our method, we treat each non-white pixel (i.e., pixels occupied by the polymer) as a "particle", using its image coordinates together with its grayscale intensity as features ($x^i \in \mathbb{R}^3$). The collection of these non-white pixels

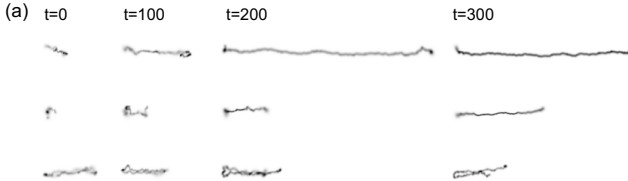

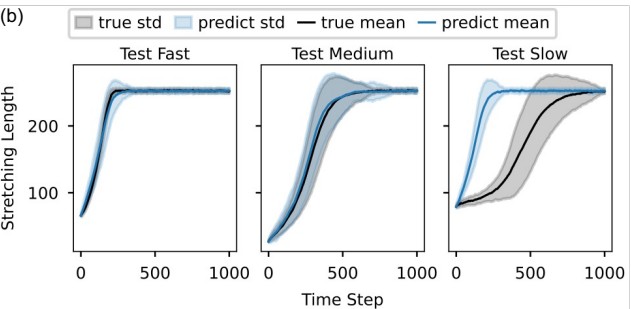

*Figure 7.* Polymer extension prediction. (a) The snapshots of the first trajectory in these three test datasets at t=0 (initial configuration), t=100, t=200 and t=300. (b) The mean and standard deviation of our predicted stretching length and the ground-truth.

defines the microstate $X$, and we compute the stretching length $\bar{z}$ from $X$. Note that the number of non-white pixels varies across time steps, and our model can deal with variable-sized microstates.

We evaluate the learned macroscopic dynamics on the three test scenarios provided in (Chen et al., 2023b), referred to as Test Fast, Test Medium, and Test Slow. Fig. 7(a) visualizes the microstates of the first trajectory at different time steps, indicating the different extension speeds. Fig. 7(b) compares our prediction to the ground-truth. It shows that our model captures the stretching-length dynamics for Test Fast and Test Medium, but overestimates the stretching rate in Test Slow. A likely reason is that the pattern of the initial configuration in Test Slow appears visually similar to that in Test Fast, making it challenging for the learned closure variable to distinguish these two. We also test the state-of-the-art image models, including CNN (Krizhevsky et al., 2012) and Vision Transformers (Dosovitskiy, 2020), to learn the latent variables by reconstructing input images. They exhibit similar results to our method, as they also succeed on Test Fast and Test Medium but fail on Test Slow. More details are provided in Appendix A.5.

## 5. Conclusion and Discussion

This work revisits a classical problem of learning macroscopic dynamics from high-dimensional microscopic observations. The novel insight delivered by it is that, instead of minimizing *pointwise reconstruction* loss, we can train an autoencoder to reconstruct the distributional informa-

tion of the input. Following this idea, we design the decoder trained to recover a smooth density representation of macrostates, making the training objective inherently permutation-invariant and allowing inputs of varying size.

A few previous works also use the decoder as the density on modeling point sets (Yang et al., 2019; Stypułkowski et al., 2021; Rasul et al., 2019). In these approaches, the observed points are treated as samples from an underlying distribution, and the decoder is trained by maximizing the (conditional) likelihood of the point set $\{\boldsymbol{x}_1, \ldots, \boldsymbol{x}_n\}$, i.e., by fitting $p_\theta(\mathbf{x}|\hat{\boldsymbol{z}})$ to the empirical samples so that the model can generate new samples. In contrast, our goal is not microstate generation, but the learning of latent closure variables. We induce, for each input set $X$, a target density $q_X$ and train the decoder by minimizing $\mathrm{KL}(q_X \| p_\theta)$. By varying $\epsilon$ in $q_X$, we can tune the resolution at which the decoder approximates.

A practical limitation is that the approach can struggle for stiff systems, where small microscopic perturbations produce large macroscopic changes. We believe this is not a limitation specific to our method, but a challenge common to all closure modeling methods. For example, consider heating a solid: throughout the process, the atoms remain close to equilibrium configurations, yet the temperature can vary substantially. The method can fail to learn a meaningful closure variable because the underlying microstates are nearly indistinguishable. In such regimes, learning a stable mapping from microstate distributions to useful closure variables may require much denser coverage of the microstate space. As a result, the current method is best suited to systems where macroscopic evolution is accompanied by substantial changes in microscopic configurations, so that the induced target distributions vary sufficiently to train the autoencoder effectively.

Several extensions are promising, including exploring other set encoders such as attention-based Set Transformers, investigating alternative density models such as score-based generative models, and improving the training strategy to couple closure modeling with dynamical prediction more tightly. We expect to apply the method to solve more real-world problems.

## Acknowledgements

This research is supported by the National Research Foundation, Singapore under its AI Singapore Programme (AISG Award No: AISG3-RP-2022-028). The computational work for this article was partially performed on resources of the National Supercomputing Centre, Singapore (https://www.nscc.sg). We thank Zhuoyuan Li and Aiqing Zhu for helpful discussions, and Fushuai Wang and Prof. Beatrice W. Soh for kindly sharing the script to generate polymer images from simulated polymer chains.

## Impact Statement

This work addresses a gap in closure modeling to learn permutation-invariant macroscopic dynamics from unordered microscopic observations. This can benefit multiscale modeling in particle-based domains, improve the scientific surrogate models and accelerate exploratory simulation and analysis. The main risk is that the prediction can become inaccurate under distribution shift or new boundary conditions that never appear in the training data. We recommend a careful out-of-distribution check when it is used for making decisions.

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

# A. Experiments Details

## A.1. Pseudocode of Train and Inference

---

**Algorithm 1** Training

---

1: **Stage I: Train the distribution-aware autoencoder.**
2: **Input:** Training trajectories $\{X_{0:T}^{(m)}\}_{m=1}^M$ where $M$ is the number of training trajectories; learnable encoder/decoder parameters $\theta$; bandwidth $\epsilon$; the number of Monte Carlo samples $K$; latent feature dimension $\hat{z}_{\text{dim}}$.
3: **Output:** Trained encoder $\hat{\varphi}$.
4: **while** stopping criterion is not met **do**
5:     Sample a minibatch of microscopic states $\mathcal{B} = \{X \mid X \in \{X_{0:T}^{(m)}\}_{m=1}^M\}$.
6:     Initialize the minibatch loss $\mathcal{L}_{\text{rec}} \leftarrow 0$.
7:     **for** each $X = \{\boldsymbol{x}^i\}_{i=1}^{|X|} \in \mathcal{B}$ **do**
8:         $\hat{\boldsymbol{z}} \leftarrow \hat{\varphi}(X)$.
9:         $q_X(\mathbf{x}) \leftarrow \frac{1}{|X|} \sum_{\boldsymbol{x}^i \in X} \delta_\epsilon(\mathbf{x} - \boldsymbol{x}^i)$.
10:        Draw Monte Carlo samples $\{\tilde{\boldsymbol{x}}^{(k)}\}_{k=1}^K \sim q_X$.
11:        $\widehat{\mathcal{L}}_{\text{rec}}(X) \leftarrow \frac{1}{K} \sum_{k=1}^K \left[ \log q_X(\tilde{\boldsymbol{x}}^{(k)}) - \log p_\theta(\tilde{\boldsymbol{x}}^{(k)} \mid \hat{\boldsymbol{z}}) \right]$.
12:        $\mathcal{L}_{\text{rec}} \leftarrow \mathcal{L}_{\text{rec}} + \widehat{\mathcal{L}}_{\text{rec}}(X)$.
13:     **end for**
14:     $\theta \leftarrow \theta - \alpha \nabla_\theta \frac{1}{|\mathcal{B}|} \mathcal{L}_{\text{rec}}$.
15: **end while**
16:
17: **Stage II: Train the dynamics model on the augmented latent state $\boldsymbol{z}_t = [\bar{\boldsymbol{z}}_t, \hat{\boldsymbol{z}}_t]$, where $\bar{\boldsymbol{z}}_t = \bar{\varphi}(X_t)$ and $\hat{\boldsymbol{z}}_t = \hat{\varphi}(X_t)$.**
18: **Input:** Training trajectories $\{X_{0:T}^{(m)}\}_{m=1}^M$; deterministic macro extractor $\bar{\varphi}$; trained encoder $\hat{\varphi}$; dynamics model $(\boldsymbol{g}, \boldsymbol{\Sigma})$ with learnable parameters $\eta$.
19: **Output:** Trained dynamics model $(\boldsymbol{g}, \boldsymbol{\Sigma})$.
20: **for** each training trajectory $X_{0:T} \in \{X_{0:T}^{(m)}\}_{m=1}^M$ **do**
21:     **for** $t = 0$ to $T$ **do**
22:         $\hat{\boldsymbol{z}}_t \leftarrow \hat{\varphi}(X_t)$.
23:         $\bar{\boldsymbol{z}}_t \leftarrow \bar{\varphi}(X_t)$.
24:         $\boldsymbol{z}_t \leftarrow [\bar{\boldsymbol{z}}_t, \hat{\boldsymbol{z}}_t]$.
25:     **end for**
26: **end for**
27: **while** stopping criterion is not met **do**
28:     Sample a minibatch of one-step pairs $\{(\boldsymbol{z}_t, \boldsymbol{z}_{t+1})\}$.
29:     **if** stochastic dynamics **then**
30:         $p_\eta(\boldsymbol{z}_{t+1} \mid \boldsymbol{z}_t) = \mathcal{N}\big(\boldsymbol{z}_t + \boldsymbol{g}_\eta(\boldsymbol{z}_t)\Delta t, \ \Delta t \, \boldsymbol{\Sigma}_\eta(\boldsymbol{z}_t)\boldsymbol{\Sigma}_\eta(\boldsymbol{z}_t)^\top\big)$.
31:        Calculate $\mathcal{L}_{\text{dyn}}$ as $\mathbb{E}_{(\boldsymbol{z}_t, \boldsymbol{z}_{t+1})} \left[ -\log p_\eta(\boldsymbol{z}_{t+1} \mid \boldsymbol{z}_t) \right]$ with the sampled minibatch data.
32:     **else**
33:        $\boldsymbol{\Sigma}_\eta \equiv 0$.
34:        Calculate $\mathcal{L}_{\text{dyn}}$ as $\mathbb{E}_{(\boldsymbol{z}_t, \boldsymbol{z}_{t+1})} \left[ \|\boldsymbol{z}_{t+1} - (\boldsymbol{z}_t + \boldsymbol{g}_\eta(\boldsymbol{z}_t)\Delta t)\|_2^2 \right]$ with the sampled minibatch data.
35:     **end if**
36:     $\eta \leftarrow \eta - \alpha \nabla_\eta \mathcal{L}_{\text{dyn}}$.
37: **end while**

---

## A.2. Evaluation Metrics

For the ODE dynamics, the evaluation metric is the mean relative error (MRE) which is defined as:

$$\text{MRE}(\mathcal{T}_{\text{test}}) = \frac{1}{|\mathcal{T}_{\text{test}}|} \sum_{\mathcal{T}_{\text{test}}} \left( \frac{\sum_t \|\bar{\boldsymbol{z}}_t^{\text{true}} - \bar{\boldsymbol{z}}_t^{\text{pred}}\|_2^2}{\sum_t \|\bar{\boldsymbol{z}}_t^{\text{true}}\|_2^2} \right),$$

---

**Algorithm 2** Inference / Rollout Prediction

---

1: **Input:** Initial microscopic configuration $X_0$; trained encoder $\hat{\varphi}$; deterministic macro extractor $\bar{\varphi}$; trained dynamics model $(g, \Sigma)$.
2: **Output:** Macrostate rollout $\{\bar{z}_0, \bar{z}_1, \ldots, \bar{z}_H\}$.
3: $\bar{z}_0 \leftarrow \bar{\varphi}(X_0)$.
4: $\hat{z}_0 \leftarrow \hat{\varphi}(X_0)$.
5: $z_0 \leftarrow [\bar{z}_0, \hat{z}_0]$.
6: **for** $t = 0$ to $H - 1$ **do**
7:   **if** stochastic dynamics **then**
8:     $z_{t+1} \sim \mathcal{N}\big(z_t + g(z_t)\Delta t, \ \Delta t\, \Sigma(z_t)\Sigma(z_t)^\top\big)$.
9:   **else**
10:     $z_{t+1} \leftarrow z_t + g(z_t)\Delta t$.
11:   **end if**
12:   Parse $z_{t+1}$ as $z_{t+1} = [\bar{z}_{t+1}, \hat{z}_{t+1}]$.
13: **end for**
14: **return** $\{\bar{z}_t\}_{t=0}^H$.

---

where $\mathcal{T}_{\text{test}}$ represents the trajectories in the test data and $|\mathcal{T}_{\text{test}}|$ is the number of test trajectories. $\{\bar{z}_t^{\text{pred}}\}_{t=1}^T$ and $\{\bar{z}_t^{\text{true}}\}_{t=1}^T$ denote the predicted and ground-truth states for a testing trajectory.

For the SDE macrodynamics, we also predict the whole trajectory for each initial state autoregressively in the testing data and evaluate the discrepancy between the predicted and true trajectory distributions using the time-average multi-kernel maximum mean discrepancy (MMD). Again, let $\{\bar{z}_t^{\text{pred}}\}_{t=1}^T$ and $\{\bar{z}_t^{\text{true}}\}_{t=1}^T$ denote the predicted and true states, and let $\bar{z}_t^{\text{pred},(i)}$ and $\bar{z}_t^{\text{true},(i)}$ be the $i$-th trajectory in $\mathcal{T}_{\text{test}}$ (suppose $|\mathcal{T}_{\text{test}}| = N$) at time $t$. The per-timestep multi-kernel MMD is computed as

$$\widehat{\text{MMD}}_t^2 = \frac{1}{|\Gamma|} \sum_{\sigma \in \Gamma} \left[ \frac{1}{N^2} \sum_{i=1}^N \sum_{j=1}^N k_\sigma\Big(\bar{z}_t^{\text{pred},(i)}, \bar{z}_t^{\text{pred},(j)}\Big) + \frac{1}{N^2} \sum_{i=1}^N \sum_{j=1}^N k_\sigma\Big(\bar{z}_t^{\text{true},(i)}, \bar{z}_t^{\text{true},(j)}\Big) \right.$$
$$\left. - \frac{2}{N^2} \sum_{i=1}^N \sum_{j=1}^N k_\sigma\Big(\bar{z}_t^{\text{pred},(i)}, \bar{z}_t^{\text{true},(j)}\Big) \right],$$

where the RBF kernel is

$$k_\sigma(\boldsymbol{a}, \boldsymbol{b}) = \exp\left( -\frac{\|\boldsymbol{a} - \boldsymbol{b}\|_2^2}{2\sigma^2} \right), \qquad \Gamma = \{0.01, 0.05, 0.25, 1.25\}.$$

Finally, averaging over timesteps and test trajectories yields

$$\text{MMD}(\mathcal{T}_{\text{test}}) = \frac{1}{|\mathcal{T}_{\text{test}}|} \sum_{\mathcal{T}_{\text{test}}} \left( \frac{1}{T} \sum_{t=1}^T \widehat{\text{MMD}}_t^2 \right).$$

### A.3. Interaction Particle System

The total pairwise energy of the interaction particle studied in Sec. 4.2 is computed as

$$E_{\text{total}} = \sum_{i<j} \left[ \frac{1}{a} \log \cosh\big(a(1 - r_{ij})\big) + b(1 - r_{ij}) \right],$$

where $r_{ij}$ is the distance between particle $i$ and particle $j$. We set $a = 4$ and $b = 0.1$. We divide this total energy by $n \times (n - 1)/2$ to get the normalized pairwise interaction energy.

### A.4. Binary Particle Mixing

Because the numbers of type-A and type-B particles vary across trajectories, baselines with an MLP decoder cannot reconstruct each species separately. Therefore, we train AE-Aug, AE-InvE, and AE-InvE-CD to reconstruct the states of all

particles jointly, without conditioning on type. For baselines (AE-InvE, AE-InvE-CD and InvE) that can incorporate type information, we use a *shared* encoder applied to each type and aggregate the resulting embeddings into a single closure variable, matching the setup of our method for a fair comparison. The encoder of baseline AE-Aug just takes all particles as the input to avoid the varying numbers of type-A and type-B particles.

### A.5. Polymer Image Dataset

The original polymer dataset is from (Chen et al., 2023b). In detail, they simulate the dynamics of polymer chains using Brownian dynamics approach in a planar elongational flow. Each polymer chain consisted of 300 beads moving in (x,y,z) dimensions. Every trajectory contains 1001 time steps. They provide 610 trajectories for training, 110 trajectories for validation and three testing cases ("test fast", "test medium" and "test slow"). In each testing case, they simulate 100 trajectories of the same initial configuration of the 300 beads.

Given the polymer simulation, we construct the video dataset in the following way. Each polymer frame (3D bead coordinates) is projected into a 2D image by placing a Gaussian blob at each bead's $(x, y)$ position on a fixed 500×100 grid. The Gaussian width depends on the bead's $z$-offset from the frame's mean $z$ (larger $|z|$, wider blob), then all blobs are summed, normalized to the frame's max intensity, and quantized to 8-bit grayscale. Fig.8 visualizes an arbitrary trajectory in the video dataset.

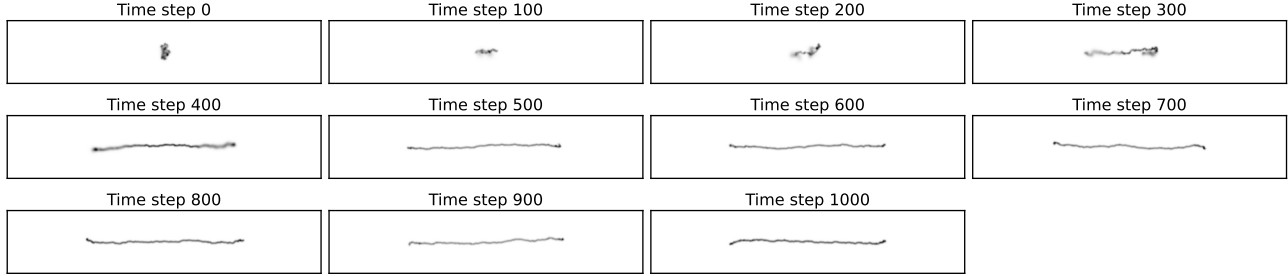

*Figure 8.* Exemplary trajectory in the polymer video dataset.

We test the state-of-the-art image models, CNN and ViT, for learning the closure variables. We train CNN and ViT autoencoders on image reconstruction with MSE loss and use the latent vector as the closure variables. The macroscopic dynamics learned with their closure variables are shown in Fig. 9.

## B. Model Implementation Details

### B.1. Encoder

The encoder implements $\hat{\boldsymbol{\varphi}} : \mathcal{X} \to \hat{\mathcal{Z}}$ (Sec. 3.2). We choose DeepSet (Zaheer et al., 2017) but any permutation-invariant architecture can also be the choice. In detail, the elementwise function $\phi$ is applied on the input feature of every particle and outputs a latent representation. We use mean pooling to aggregates the latent representations to the set-level representation. Finally, the set-level neural network $\rho$ transfers the aggregated set-level representation to the final $\hat{\boldsymbol{z}}$. We implement the elementwise function $\phi$ and the set-level function $\rho$ by MLPs.

### B.2. Decoder

The decoder $\boldsymbol{\psi}$ parameterizes a conditional density $p(\mathbf{x}|\hat{\boldsymbol{z}})$, i.e., $p(\mathbf{x}|\hat{\boldsymbol{z}}) = \boldsymbol{\varphi}(\mathbf{x}; \hat{\boldsymbol{z}})$. We instantiate $p(\mathbf{x}|\hat{\boldsymbol{z}})$ with a stack of conditional autoregressive rational quadratic spline flow layers (Durkan et al., 2019) since it is powerful to approximate complex densities and allows us to compute the likelihood conveniently. But any conditional density model is applicable. In detail, the conditional normalizing flow defines an invertible mapping $\mathbf{x} = f(\mathbf{u}; \hat{\boldsymbol{z}})$ from a simple base random variable $\mathbf{u}$ (e.g., $\mathbf{u} \sim \mathcal{N}(\mathbf{0}, \mathbf{I})$) to the particle feature $\mathbf{x}$. By the change-of-variables formula, the conditional log-likelihood $\log p(\mathbf{x}|\hat{\boldsymbol{z}})$ can be computed exactly, allowing for training by maximizing likelihood.

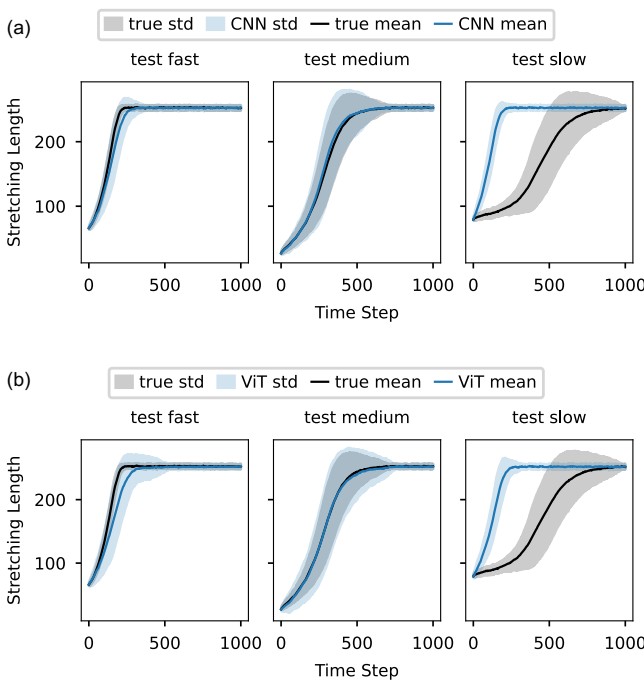

*Figure 9.* Polymer extension prediction with the closure variables learned by (a) CNN and (b) ViT.

### B.3. The Induced Density

The $q_X(\mathbf{x})$ constructed in Sec. 3.2 can be viewed as a kernel density estimate of the empirical measure associated with $X$. As $\epsilon$ goes close to 0 (with $|X|$ fixed), $q_X$ concentrates and converges in the weak sense to the empirical measure $\frac{1}{|X|}\sum_j \delta_{\boldsymbol{x}_j}$ (i.e., a sum of point masses) (Silverman, 2018). When the decoder family $p(\cdot|\hat{\boldsymbol{z}})$ is restricted to smooth densities (e.g., a normalizing flow), we keep $\epsilon > 0$ and interpret $p(\cdot|\hat{\boldsymbol{z}}) \approx q_X$ as *distributional recovery at resolution $\epsilon$*, rather than exact pointwise reconstruction of every particle in the limit $\epsilon \to 0$.

## C. Supplementary Numerical Results

### C.1. Compatibility with Encoders Beyond DeepSet

To demonstrate that the proposed framework is not restricted to DeepSet, we replaced the DeepSet encoder with a Set Transformer (Lee et al., 2019) in the interacting particle system experiment while keeping all other settings unchanged. The results are reported in Table 3.

*Table 3.* Mean relative rollout prediction error for different encoder architectures in the interacting particle system. The mean and standard deviation are computed from three runs.

|  | in-dst | diff-init | diff-N |
|---|---|---|---|
| DeepSet | $5.19_{\pm 0.25}\times 10^{-5}$ | $\mathbf{4.38}_{\pm\mathbf{0.19}}\times 10^{-4}$ | $5.22_{\pm 0.46}\times 10^{-5}$ |
| Set Transformer | $\mathbf{3.20}_{\pm\mathbf{0.48}}\times 10^{-5}$ | $1.70_{\pm 0.24}\times 10^{-3}$ | $\mathbf{2.60}_{\pm\mathbf{0.23}}\times 10^{-5}$ |

The Set Transformer achieves lower rollout prediction errors in the in-dst and diff-N settings, whereas DeepSet is better under the diff-init distribution shift setting. These results suggest that more expressive permutation-equivariant encoders can improve performance when the testing distribution is close to the training, but may also learn more specialized latent representations that are less robust under distribution shift. Overall, the experiment empirically demonstrates that the proposed framework is compatible with different encoders and that the main contribution is orthogonal to the encoder choice.

**C.2. Sensitivity to the Latent Dimension $\hat{z}_{\mathrm{dim}}$ and Smoothing Scale $\epsilon$**

In general, closure modeling aims to compress the high-dimensional microscopic information into a low-dimensional latent variable, which can be used as the state vector for modeling the system's macroscopic dynamics. In our method, the latent dimension $\hat{z}_{\mathrm{dim}}$ and smoothing scale $\epsilon$ both affect the information compression, i.e., how much microscopic information is encoded into the closure variable. Specifically, $\hat{z}_{\mathrm{dim}}$ controls the information compression capacity, while $\epsilon$ controls the information compression resolution. A larger $\epsilon$ smooths out fine-scale details, whereas a smaller $\epsilon$ preserves detailed information.

To study their influence, we perform a sensitivity analysis on the in-dst setting of the interacting particle system. Table 4 shows that the rollout prediction error decreases as the latent dimension increases and becomes stable once $z^{\mathrm{dim}} \geq 5$, suggesting that the latent representation has sufficient capacity to encode the microscopic information needed for modeling macroscopic dynamics beyond this point. The large error with small $\hat{z}_{\mathrm{dim}}$ (e.g., $\hat{z}_{\mathrm{dim}} = 1$) is under expectation because the too small $\hat{z}_{\mathrm{dim}}$ poses an overly restrictive bottleneck to encode microscopic information.

Table 5 shows that the method remains relatively stable across a broad range of smoothing scales. Interestingly, even extremely small values of $\epsilon$ (e.g., $10^{-8}$) still produce useful closure variables for macroscopic dynamics prediction. The reason is as follows. In the limit $\epsilon \to 0$, the reconstruction objective (Eq. 3) effectively reduces to maximizing likelihood on the observed point set. In this regime, the learned density no longer accurately reconstructs the smoothed target density, but still preserves sufficient underlying distributional structure of the particles, which remains informative for predicting macroscopic dynamics.

*Table 4.* Sensitivity analysis of the latent dimension $\hat{z}_{\mathrm{dim}}$ on the rollout prediction error in the in-dst setting of the interacting particle system. The mean and standard deviation are computed from three runs.

| $\hat{z}_{\mathrm{dim}}$ | 1 | 2 | 3 | 4 | 5 | 6 | 7 | 8 |
|---|---|---|---|---|---|---|---|---|
| mean | $1.6 \times 10^{-3}$ | $1.1 \times 10^{-3}$ | $7.0 \times 10^{-4}$ | $7.2 \times 10^{-4}$ | $4.5 \times 10^{-5}$ | $4.6 \times 10^{-5}$ | $5.0 \times 10^{-5}$ | $5.2 \times 10^{-5}$ |
| std | $8.6 \times 10^{-6}$ | $4.5 \times 10^{-4}$ | $3.0 \times 10^{-5}$ | $4.1 \times 10^{-5}$ | $1.5 \times 10^{-6}$ | $2.3 \times 10^{-6}$ | $1.9 \times 10^{-6}$ | $2.5 \times 10^{-6}$ |

*Table 5.* Sensitivity analysis of the smoothing scale $\epsilon$ on the rollout prediction error in the in-dst setting of the interacting particle system. The mean and standard deviation are computed from three runs.

| $\epsilon$ | $10^{-1}$ | $10^{-2}$ | $10^{-3}$ | $10^{-4}$ | $10^{-5}$ | $10^{-6}$ | $10^{-7}$ | $10^{-8}$ |
|---|---|---|---|---|---|---|---|---|
| mean | $6.9 \times 10^{-5}$ | $5.2 \times 10^{-5}$ | $5.0 \times 10^{-5}$ | $5.5 \times 10^{-5}$ | $5.5 \times 10^{-5}$ | $6.4 \times 10^{-5}$ | $5.3 \times 10^{-5}$ | $6.1 \times 10^{-5}$ |
| std | $4.3 \times 10^{-6}$ | $2.5 \times 10^{-6}$ | $1.2 \times 10^{-5}$ | $2.7 \times 10^{-6}$ | $9.7 \times 10^{-6}$ | $9.0 \times 10^{-6}$ | $3.8 \times 10^{-6}$ | $4.0 \times 10^{-6}$ |

