# OpenReview forum: "Learning Permutation-invariant Macroscopic Dynamics"
_ICML.cc/2026/Conference — ICML 2026 regular_

### Official Review · Reviewer_jjAj · 2026-03-05

**Soundness:** 3
**Presentation:** 3
**Significance:** 3
**Originality:** 3
**Overall Recommendation:** 5
**Confidence:** 4

**Summary:**

This paper introduces a method for learning macroscopic dynamics from unordered microscopic observations. Instead of using traditional point-to-point autoencoders, the authors propose a "distribution reconstruction" autoencoder that learns to reconstruct the kernel density (KDE) induced by the point set, avoiding issues with point ordering. The approach uses a permutation-invariant encoder (DeepSet) and conditional normalizing flows for density decoding. The macroscopic dynamics are modeled using ODE/SDEs. Experiments cover particle systems and polymer dynamics, showing the effectiveness of the method in handling unordered data and achieving accurate macroscopic predictions.

**Compliance With Llm Reviewing Policy:**

Affirmed.

**Key Questions For Authors:**

1. Could the authors provide the impact of 𝜖 on downstream prediction tasks, rather than only the distribution reconstruction task shown in Figure 4?

2. Could the authors further analyze the failure modes observed in the polymer expansion task? The authors mention that the closure variable may fail to distinguish the pattern of the initial configuration. However, even CNN and ViT also failed on this task. Can the authors analyze whether the failure of the visual model lies at the level of visual representation or at the level of downstream dynamic simulation?

3. Could the authors show how the system size affects the training time of the conditional flow model?

**Limitations:**

yes

**Strengths And Weaknesses:**

**Strengths**

Clear core idea and problem relevance: Replacing point-to-point reconstruction with distribution-based reconstruction effectively addresses the challenges of unordered point sets in dynamic systems.

Direct validation of permutation invariance: The authors show that their model maintains invariant predictions under various permutations, whereas traditional methods fail due to dependence on specific point orderings.

Coverage of "changing particle numbers/distribution shift": The authors include tests for particle number variation (diff-N) and initialization distribution variation (diff-init), demonstrating robust performance against distribution shifts in their experiments.

Comprehensive baseline comparisons: The paper compares their approach with several other methods, such as permutation augmentation + MLP AE (AE-Aug), permutation-invariant encoder but point-to-point MSE decoding (AE-InvE), and Chamfer reconstruction (AE-InvE-CD), providing valuable insights into the benefits of distribution reconstruction.

**Weaknesses**
Lack of clarity on choice and sensitivity of key hyperparameter 𝜖 (KDE bandwidth): The selection of 𝜖 is not fully explained. This hyperparameter is sensitive to both particle resolution and closure variable identifiability, and the paper could benefit from a more detailed analysis of how different values of 𝜖 affect macroscopic prediction stability.

Experimental results not always clearly superior, with notable failure cases: In the interacting particle experiment (diff-init), AE-InvE-CD outperforms the proposed method, which is described as "competitive" but not optimal. The polymer expansion task shows overestimated elongation rates in the Test Slow scenario, suggesting that the current model might not fully address the challenges of this dataset.

Application boundaries and computational cost: The authors claim that the cost of KL estimation is dominated by the MC sample count rather than particle number. However, training the conditional flow model may still be computationally expensive, especially for high-dimensional particle features.

---

> ### Author Rebuttal · Authors · 2026-03-31
>
> We thank the reviewer for the careful reading and constructive comments. We address the questions below.
>
> **Impact of $\epsilon$ on downstream prediction tasks.**
>
> Conceptually, the smoothing scale $\epsilon$ controls the resolution of the target density. If $\epsilon$ is too large, fine-scale information is oversmoothed; if it is too small, the target density becomes highly concentrated, which makes the distributional reconstruction difficult. We empirically evaluate the effect of $\epsilon$ on rollout prediction in the interacting particle system (mean and standard deviation over three runs). The results are as follows:
>
> |  | $10^{-1}$ | $10^{-2}$ | $10^{-3}$ | $10^{-4}$ | $10^{-5}$ | $10^{-6}$ | $10^{-7}$ | $10^{-8}$ |
> |---|---:|---:|---:|---:|---:|---:|---:|---:|
> | mean | $6.9 \times 10^{-5}$ | $5.2 \times 10^{-5}$ | $5.0 \times 10^{-5}$ | $5.5 \times 10^{-5}$ | $5.5 \times 10^{-5}$ | $6.4 \times 10^{-5}$ | $5.3 \times 10^{-5}$ | $6.1 \times 10^{-5}$ |
> | std | $4.3 \times 10^{-6}$ | $2.5 \times 10^{-6}$ | $1.2 \times 10^{-5}$ | $2.7 \times 10^{-6}$ | $9.7 \times 10^{-6}$ | $9.0 \times 10^{-6}$ | $3.8 \times 10^{-6}$ | $4.0 \times 10^{-6}$ |
>
> The rollout prediction performance is fairly stable across a broad range of $\epsilon$. Interestingly, even a very small $\epsilon$ (e.g., $10^{-8}$) still yields useful closure variables for macro dynamics modeling. The reason is as follows. In the limiting case $\epsilon \to 0$, the Monte Carlo samples in Eq.~(3) reduce to the particles themselves, so the objective effectively becomes maximum likelihood on the observed point set. We verified this by comparing the plots of target density and the learned density. In this regime, the learned density no longer accurately reconstructs the smoothed target density, but it can still capture some underlying distributional structure of the particles, which remains informative for predicting macroscopic dynamics.
> We will include this experiment in the revision.
>
> **Failure modes in the polymer expansion task**
>
> We believe the failure is due to the visual representation. The reason is as follows. The original polymer dataset comes from MD simulation in [1]. One important thing here is that the order of beads in a polymer matters because the order reflects how the beads are connected. The connectivity, together with the beads' 3D positions, determines the dynamics of the polymer. Therefore, a suitable closure variable should capture not only bead positions, but also how the beads are connected. However, in our experiment, the closure variable is learned from the visual representation obtained by projecting each 3D polymer configuration onto a 2D grid with Gaussian rendering. This means that the visual representation does not explicitly preserve the bead connectivity and also loses part of the positional information. As a result, learning a closure variable to summarize the underlying microstates, including beads positions and connectivity, is extremely difficult in the polymer experiment. In particular, the initial images in Test Slow and Test Fast appear visually similar, which likely explains why both our model and state-of-the-art vision baselines fail. We also emphasize that our method is not specifically designed for such image input because the image pixels already have a canonical ordering. The purpose of this experiment is to demonstrate that our method can also be applied to alternative input formats and achieve performance comparable to feature extractors designed specifically for images, not just the raw particle coordinates in MD simulation.
>
> **Effect of system size on conditional-flow training time.**
>
> Our current implementation uses the conditional autoregressive rational-quadratic spline flow as the decoder. Since the reconstruction loss is computed by estimating the KL divergence using Monte Carlo (MC) samples drawn from the target distribution, the computational cost of the decoder depends on the number of MC samples, the input feature dimension, and the cost of autoregressive flow evaluations, rather than the number of input particles. This is one advantage over pointwise reconstruction losses because our method avoids the point set matching, which is expensive when the number of particles is large. As for the optimization stability, we know the Monte Carlo estimator is unbiased. But using too few samples can introduce substantial variance in the reconstruction objective. Therefore, we should increase the MC samples when the training is unstable. We will clarify the computational complexity in the revision.
>
> **Reference:**
>
> [1] Chen, Xiaoli, et al. "Constructing custom thermodynamics using deep learning." Nature Computational Science 4.1 (2024): 66-85.

---

> > ### Author Rebuttal · Reviewer_jjAj · 2026-04-01
> >
> > We thank the authors for the detailed rebuttal; the reviewer's concerns have been addressed, and the rating will be adjusted to "Accept." Additionally, there is a question unrelated to the score that I hope the authors can address (since the proposed method is not specifically designed for such image input). Regarding failure modes in the polymer expansion task: what type of visual representation is capable of simultaneously capturing both the positional information of the beads and the connectivity relationships between them? Would advanced pre-trained visual representations—such as DINOv3—or object-centric models like Slot Attention be suitable for this purpose?

---

> > > ### Author Response · Authors · 2026-04-02
> > >
> > > We thank the reviewer for the positive assessment and thoughtful comment.
> > > Actually, it is a good question. We think it is difficult to learn a good encoding that captures both positional information and the connectivity from self-supervised image reconstruction alone, where the image only explicitly shows the beads. The reason is as follows. A purely image-based representation may well capture bead positions since beads are directly reflected in the reconstructed image. But the connectivity is not explicitly observable from pixels and is therefore inferred from spatial patterns of beads. This is very challenging for many configurations, especially when beads are close to each other. Therefore, a potential solution is to also record the connectivity information in the image explicitly. For example, such information could in principle be made more explicit through advanced labeling strategies in experiments, although doing so in practice would remain challenging [1].
> > >
> > > Regarding the advanced pre-trained visual representations, we think it is a very promising direction. We could use these pre-trained visual representations directly or even fine-tune them on the images from advanced experimental techniques that make connectivity-related information more explicit (e.g., fine-tuning to project the pre-trained visual representations to a low-dimensional latent space, which is effective for learning macroscopic dynamics). We view this as a promising future work specifically targeted at closure modeling for image inputs.
> > >
> > >
> > >
> > > **Reference:**
> > >
> > > [1] Mai, Danielle J., and Charles M. Schroeder. "100th anniversary of macromolecular science viewpoint: single-molecule studies of synthetic polymers." ACS Macro Letters 9.9 (2020): 1332-1341.

---

### Official Review · Reviewer_QMdA · 2026-03-14

**Soundness:** 3
**Presentation:** 3
**Significance:** 3
**Originality:** 4
**Overall Recommendation:** 5
**Confidence:** 3

**Summary:**

This paper studies closure modeling when the microscopic state is an unordered set, such as a particle system without a canonical indexing. The method uses a permutation-invariant encoder to map the set-valued microstate into a latent closure variable, and replaces pointwise reconstruction with distributional reconstruction: instead of reconstructing particle coordinates directly, the decoder learns a conditional density that matches a smoothed empirical distribution induced by the observed particles. The latent closure variable is then concatenated with target macroscopic observables and evolved with a learned deterministic or stochastic reduced dynamics model. The paper evaluates this framework on interacting particle systems, Lennard–Jones mixing, and polymer-image data, and argues that the approach is exactly permutation-invariant and robust across varying particle counts.

**Compliance With Llm Reviewing Policy:**

Affirmed.

**Final Justification:**

Thank you for the rebuttal and follow-up clarification. I continue to view the paper very positively. The paper addresses an important problem and proposes a clear, technically solid, and meaningfully original solution based on learning permutation-invariant closure variables through distribution reconstruction. The rebuttal clarified the intended scope of the method, sharpened its distinction from related point-cloud and density-modeling approaches, and satisfactorily addressed my concern about sensitivity to the smoothing scale and latent dimension through the follow-up quantitative analysis. Overall, the rebuttal reinforced my original assessment, and I maintain my recommendation.

**Key Questions For Authors:**

1 The method is very natural for unordered particle systems, but how broadly do the authors expect it to generalize beyond this setting? In particular, could the authors clarify the intended scope for microstates with partial ordering or grid-based structure? My assessment of the paper’s generality would depend on this clarification.

2 The smoothing scale 𝜖 and the latent dimension Zdim appear to be important hyperparameters. Could the authors provide a more systematic sensitivity analysis showing how these choices affect final rollout prediction performance in the main experiments? This would help clarify practical robustness and reproducibility.

3 In the interacting-particle diff-init setting, AE-InvE-CD performs best, and in the polymer Test Slow setting the proposed method fails similarly to image-based baselines. What regimes do the authors believe are the true strengths of the proposed method? Beyond “unorderedness,” how important are factors such as the amount of microscopic distributional change or the stiffness of the system? My evaluation of the paper’s scope would depend on this answer.

4 Relative to point-cloud reconstruction baselines, is the main benefit of the proposed method primarily that it avoids explicit matching, or that it represents only the information needed for closure modeling through a density, or both? Clarifying this point would make the novelty relative to related work easier to assess.

**Limitations:**

yes

**Strengths And Weaknesses:**

This paper addresses an important and natural problem: learning macroscopic dynamics from microscopic states that do not have a canonical ordering. The main idea is clear and well motivated. Instead of relying on point-wise reconstruction, which implicitly assumes indexed microscopic degrees of freedom, the paper proposes learning permutation-invariant closure variables through distributional reconstruction. The overall pipeline is coherent, combining a DeepSet-based permutation-invariant encoder, a conditional density decoder, and a latent dynamical model over the macroscopic observables and learned closure variables.

From a soundness perspective, the empirical evidence is generally convincing. In the interacting-particle experiment, the paper directly verifies permutation invariance, which is particularly valuable because it tests the central claim of the method rather than only reporting prediction error. The evaluation also includes shifts in particle number and initial configuration, which strengthens the experimental design. In addition, the method achieves the best MMD across all reported settings in the binary Lennard–Jones mixing experiment, suggesting that it is effective at least for particle-based unordered microstates.

That said, the strengths should be calibrated against several limitations. First, the advantage is not uniform across all settings. In the interacting-particle diff-init setting, AE-InvE-CD performs best, and in the polymer experiment the proposed method fails on Test Slow; moreover, CNN/ViT baselines show qualitatively similar behavior there. This suggests that the method is especially compelling for unordered particle-like microstates, rather than uniformly superior across all modalities. Second, the results indicate sensitivity to the smoothing scale 𝜖 and latent dimension Zdim, but the paper does not yet provide a systematic analysis of how these design choices affect downstream rollout quality in the main experiments. Third, while the paper is generally well written, the distinction from point-cloud reconstruction and density-based generative modeling could be sharpened further. Since the paper’s goal is closure-variable learning rather than sample generation, clarifying this contrast more explicitly would strengthen the originality claim.

Overall, I find the paper technically solid and meaningful. The originality comes less from introducing an entirely new building block and more from reformulating closure modeling for unordered microstates in a principled way. The impact may be somewhat specialized rather than broad, but within that scope the contribution appears significant and likely useful for future work on multiscale modeling from set-valued microscopic observations.

---

> ### Author Rebuttal · Authors · 2026-03-31
>
> We thank the reviewer for the thoughtful and valuable feedback. We address the concerns below and will incorporate the clarifications in the revision.
>
> **Distinction from point-cloud reconstruction and density-based generative modeling.**
>
> We appreciate that the reviewer identified this important point. Our goal is *not* sample generation or accurate recovery of the microstate for its own sake. Rather, the goal is to learn a permutation-invariant latent closure variable that summarizes the microscopic information. We use the conditional normalizing flow as a decoder to train the latent variable to retain distributional information rather than density-based generative modeling.
>
> **Scope beyond unordered particle systems.**
>
> In principle, the method applies when the high-dimensional observations can be modeled as evolving according to some approximately Markovian dynamics. In this sense, it is also applicable to partially ordered or grid-based data, provided their dynamics can be treated as approximately Markovian. For example, we tested our method on a PDE system on a 2D grid used in [1] and observed that it works well there. That said, our claim is not that the proposed approach should replace structure-aware models for all grid-based data. Rather, we view it as a solution in the challenging regimes for previous closure modeling methods, where unorderedness and varying input size are intrinsic to the microscopic state.
>
> **How do $\epsilon$ and latent dimension influence the dynamic prediction?**
>
> The latent dimension determines the information compression bottleneck. If it is too small, the latent variable may fail to retain sufficient information to distinguish microscopic states, which degrades downstream prediction. We provide supporting evidence for this trend in Appendix A.2. Regarding $\epsilon$, due to word limit, we refer the reviewer to our response to the same question raised by Reviewer jjAj.
>
>
> **Regimes of strength.**
>
> In our view, the proposed method is most advantageous when the microscopic state has no ordering and may have varying input sizes (i.e., different numbers of particles) across time or trajectories. In such settings, pointwise-reconstruction methods are less well aligned, even when combined with permutation-invariant losses, whereas our formulation directly learns a permutation-invariant closure variable through distribution reconstruction.
>
> Beyond "unorderedness", we believe a key factor is the *separability* of the microstates associated with different macroscopic regimes. Actually, this is not a limitation specific to our method, but a challenge common to all closure modeling methods. Specifically, closure modeling tends to succeed when macroscopic evolution is accompanied by sufficient changes in the microscopic configuration, so that microstates remain distinguishable and can support learning informative closure variables. Conversely, closure modeling is challenging when distinct macroscopic regimes correspond to nearly indistinguishable microstates. This is closely related to stiffness. In stiff systems, small microscopic differences can induce large macroscopic changes, making the closure variable difficult to infer robustly from the microscopic state. This likely contributes to the difficulty in the polymer Test Slow setting, where the initial configurations appear visually similar to Test Fast. Besides, CNN and ViT also fail there, suggesting that this is not specific to our set-based representation alone.
>
> **Main benefits compared to point-cloud reconstruction baselines.**
>
> We believe the benefit is *both*, with the second point being the more conceptual contribution.
>
> First, point-cloud reconstruction methods require comparing the reconstructed and observed sets through an explicit pointwise or matching-based objective, even when the loss is permutation-invariant. This introduces an additional alignment problem between unordered sets, which is not directly related to the closure-learning goal and can affect the learned latent representation. Our method avoids this step entirely by reconstructing a permutation-invariant target density.
>
> Second, our goal is not to reconstruct microstates as accurately as possible, but to learn the closure variable for macroscopic dynamics. Existing point-cloud reconstruction methods can in principle be used for this purpose, but they are trained to faithfully recover the point cloud itself rather than to isolate the coarse-grained information relevant for closure modeling, and therefore may preserve fine details unnecessary for closure modeling. By contrast, our method emphasizes distributional structure in the microstate rather than exact recovery of every element, which better aligns with closure modeling.
>
> **Reference:**
>
> [1] Chen, Mengyi, and Qianxiao Li. "Learning macroscopic dynamics from partial microscopic observations." Advances in Neural Information Processing Systems 37 (2024): 48996-49021.

---

> > ### Author Rebuttal · Reviewer_QMdA · 2026-04-03
> >
> > Thank you for the rebuttal. I still find the paper technically solid and the core idea—learning permutation-invariant closure variables through distribution reconstruction—interesting and meaningful. The rebuttal also clarifies the scope of the method and sharpens its distinction from point-cloud reconstruction and density-based generative modeling.
> >
> > However, one of my main questions asked for a systematic numerical sensitivity analysis of the smoothing scale and latent dimension, since these directly affect practical robustness and reproducibility. I do not think this point was sufficiently addressed with clear quantitative evidence in the rebuttal. This reduces my confidence somewhat in the empirical completeness of the paper.
> >
> > Overall, I remain positive about the paper’s core contribution, but I no longer think the evaluation supports my original, stronger score. I am therefore considering revising my recommendation downward slightly, from 5 to 4.

---

> > > ### Author Response · Authors · 2026-04-07
> > >
> > > We thank the reviewer for the helpful clarification. We realize that our previous rebuttal did not present the sensitivity analysis in a sufficiently self-contained and systematic way. We would like to clarify it below.
> > >
> > > We can think of the influence of the smoothing scale $\epsilon$ and latent dimension $\hat z_{\mathrm{dim}}$ in a unified way. In general, closure modeling aims to compress the high-dimensional microscopic information into a low-dimensional latent variable, which can be used as the state vector for modeling the system’s macroscopic dynamics. In our method, both $\hat z_{\mathrm{dim}}$ and $\epsilon$ affect the "information compression", i.e., how much microscopic information is encoded into the closure variable. In detail, $\hat z_{\mathrm{dim}}$ controls the information compression capacity. Small $\hat z_{\mathrm{dim}}$ means the latent variable can only keep little microscopic information, and vice versa. $\epsilon$ controls the information compression resolution. Large $\epsilon$ will smooth out fine-scale details to encode, whereas small $\epsilon$ preserves detailed information in the latent variable.
> > >
> > >
> > > To test the influence of $\hat z_{\mathrm{dim}}$ and $\epsilon$, we consider the *in-dst* testing case of the interacting particle systems (see Sec. 4.2 in the paper for details). The quantitative empirical results are as follows (mean and standard deviation over three runs):
> > >
> > > | $\hat z_{\mathrm{dim}}$ | 1 | 2 | 3 | 4 | 5 | 6 | 7 | 8 |
> > > |---|---:|---:|---:|---:|---:|---:|---:|---:|
> > > | mean | $1.6 \times 10^{-3}$ | $1.1 \times 10^{-3}$ | $7.0 \times 10^{-4}$ | $7.2 \times 10^{-4}$ | $4.5 \times 10^{-5}$ | $4.6 \times 10^{-5}$ | $5.0 \times 10^{-5}$ | $5.2 \times 10^{-5}$ |
> > > | std | $8.6 \times 10^{-6}$ | $4.5 \times 10^{-4}$ | $3.0 \times 10^{-5}$ | $4.1 \times 10^{-5}$ | $1.5 \times 10^{-6}$ | $2.3 \times 10^{-6}$ | $1.9 \times 10^{-6}$ | $2.5 \times 10^{-6}$ |
> > >
> > >
> > > | $\epsilon$ | $10^{-1}$ | $10^{-2}$ | $10^{-3}$ | $10^{-4}$ | $10^{-5}$ | $10^{-6}$ | $10^{-7}$ | $10^{-8}$ |
> > > |---|---:|---:|---:|---:|---:|---:|---:|---:|
> > > | mean | $6.9 \times 10^{-5}$ | $5.2 \times 10^{-5}$ | $5.0 \times 10^{-5}$ | $5.5 \times 10^{-5}$ | $5.5 \times 10^{-5}$ | $6.4 \times 10^{-5}$ | $5.3 \times 10^{-5}$ | $6.1 \times 10^{-5}$ |
> > > | std | $4.3 \times 10^{-6}$ | $2.5 \times 10^{-6}$ | $1.2 \times 10^{-5}$ | $2.7 \times 10^{-6}$ | $9.7 \times 10^{-6}$ | $9.0 \times 10^{-6}$ | $3.8 \times 10^{-6}$ | $4.0 \times 10^{-6}$ |
> > >
> > >
> > > For $\hat z_{\mathrm{dim}}$, the rollout prediction error first decreases as the latent dimension increases, and remains relatively stable when the latent dimension reaches a value (5 in this case). We checked the learned density and found that the reconstructed densities are very similar for $\hat z_{\mathrm{dim}}\geq 5$, explaining why the prediction becomes stable. The large error with small $\hat z_{\mathrm{dim}}$ (e.g., $\hat z_{\mathrm{dim}}= 1$) is under expectation because the too small $\hat z_{\mathrm{dim}}$ poses an overly restrictive bottleneck to encode microscopic information.
> > >
> > >
> > > Regarding $\epsilon$, the performance is fairly stable across a broad range. Interestingly, even a very small $\epsilon$ (e.g., $10^{-8}$) still yields useful closure variables for macro dynamics modeling. The reason is as follows. In the limiting case $\epsilon \to 0$, the Monte Carlo samples in the reconstruction objective (Eq.3 in the paper) reduce to the particles themselves, so the objective effectively becomes maximum likelihood on the observed point set. We verified this by comparing the plots of target density and the learned density. In this regime, the learned density no longer accurately reconstructs the smoothed target density, but it can still capture some underlying distributional structure of the particles, which remains informative for predicting macroscopic dynamics.
> > >
> > >
> > > We would like to thank the reviewer again for posting this question, because it makes us think of the effects of $\hat z_{\mathrm{dim}}$ and $\epsilon$ in a unified and systematic way. We would like to include this discussion in the revision.

---

### Official Review · Reviewer_q6sW · 2026-03-14

**Soundness:** 3
**Presentation:** 2
**Significance:** 3
**Originality:** 3
**Overall Recommendation:** 4
**Confidence:** 3

**Summary:**

The paper studies the problem of learning permutation-invariant macroscopic dynamics from particle-based systems. Many existing approaches learn low-dimensional closure variables using autoencoders trained for pointwise reconstruction, typically assuming a fixed ordering of microscopic degrees of freedom. However, in many settings—such as particle systems—the microscopic state is inherently unordered.

To address this, the paper proposes a two-stage framework in which a permutation-invariant autoencoder learns a latent representation of point-cloud observations that serves as a closure variable for macroscopic dynamics. The autoencoder uses a DeepSets encoder and introduces a decoder that reconstructs particle distributions via a kernel density representation and a normalizing-flow-based density model trained using KL divergence. A stochastic latent dynamics model then evolves the macroscopic observable together with the learned closure variable using an SDE trained via likelihood maximization under an Euler–Maruyama discretization.

The method is evaluated on several synthetic particle systems, including Lennard–Jones fluids, as well as on a video-based dataset modeling polymer stretching dynamics. Comparisons against a diverse set of baselines illustrate the potential advantages of the proposed approach.

**Compliance With Llm Reviewing Policy:**

Affirmed.

**Final Justification:**

The paper addresses an interesting problem—learning permutation-invariant closure variables for macroscopic dynamics—and proposes a technically appealing framework combining DeepSets representations, a density-based autoencoder, and a latent SDE model. The approach is well motivated, and the experimental evaluation is comprehensive, suggesting good soundness, originality, and potential significance.

The rebuttal addresses several of my main concerns, particularly by clarifying the training pipeline, confirming that the encoder is trained on the full set of microstates, and improving the description of the dynamics training and evaluation. These clarifications resolve important ambiguities and strengthen the paper.

However, some concerns remain only partially addressed. In particular, the connection between the density-reconstruction objective and effective closure learning is still not fully justified, and the discussion of computational complexity and optimization stability remains limited.

Overall, the rebuttal improves the paper but does not fully resolve all issues. I therefore update my assessment and raise my recommendation from weak reject to weak accept.

**Key Questions For Authors:**

(1) The proposed autoencoder introduces an interesting decoder formulation in which point clouds are represented as kernel density estimates and reconstructed via a normalizing-flow-based density model trained using a KL divergence. This is a compelling alternative to standard point-cloud reconstruction losses such as Chamfer or Earth Mover distance.

However, the learned latent representation is optimized for density reconstruction, while the downstream task requires the latent variables to act as closure variables for macroscopic dynamics. It would be helpful if the paper could discuss how well these two objectives align in practice, or whether representations optimized for density reconstruction are expected to capture the latent variables most relevant for the dynamical evolution.

(2) The proposed framework constructs closure variables by applying the encoder to point-cloud representations of the microscopic state $X_t$. While the paper demonstrates that such point clouds can be obtained either directly from particle simulations or indirectly from images (e.g., by extracting non-white pixels in the video experiment), the approach still assumes access to particle-level observations at each time step. In many real-world settings only macroscopic or aggregated measurements are available, in which case computing such closure variables may not be straightforward. A brief discussion of the applicability of the method in settings where only coarse or macroscopic observations are available would help clarify the scope of the approach.

**Limitations:**

The authors discuss several practical limitations of the proposed framework in the manuscript, including potential difficulties when modeling stiff systems and the challenge of learning meaningful closure variables when the underlying microstates exhibit only limited variability. These discussions help clarify the regimes in which the method is expected to perform well.

However, it would be helpful to further discuss how the proposed density-based decoder affects computational complexity and optimization stability, particularly when scaling to larger particle systems or higher-dimensional point clouds.

The included impact statement addresses several relevant aspects of potential broader impacts.

**Strengths And Weaknesses:**

### Strengths

- The paper proposes an interesting decoder formulation for point-cloud autoencoders in which the point cloud is represented as a kernel density estimate and reconstructed via a normalizing-flow-based density model trained using KL divergence. This avoids point-matching losses such as Chamfer or Earth Mover distance and naturally handles varying numbers of particles.

- The framework combines permutation-invariant representations (DeepSets) with a latent SDE model to learn macroscopic dynamics from particle-based systems. This design is well motivated for modeling stochastic physical systems where the number of particles may vary over time.

- The experimental evaluation is comprehensive. The authors evaluate the proposed framework on several synthetic particle systems and real-world data. The experiments include analyses of the distributional reconstruction capabilities of the proposed autoencoder (e.g., ablations of the kernel density variance), comparisons against a diverse set of baseline methods demonstrate improved performance on predicting mixing dynamics in Lennard–Jones fluids, and an experiment on video data modeling polymer stretching dynamics. The release of a code library further supports the reproducibility of the results.

- The paper is well structured, and the figures help illustrate the architecture of the proposed framework and its main components.

### Weaknesses

- **Clarity of the training procedure.** The description of the training pipeline is somewhat difficult to follow. The paper introduces a combined objective $L = L_{rec} + \lambda_{dyn} L_{dyn}$, which suggests joint optimization of the representation and the dynamical model. However, the text later states that the latent states and the dynamical model are learned separately and that the autoencoder is trained only on $X_0$. From the description and Figure 3, it appears that the encoder is first trained independently and then used to compute latent representations (closure variables $\hat z_t$) from the microscopic states $X_t$ at each time step. These encoded variables are then concatenated with the macroscopic observable $\overline z_t$, and the SDE dynamics model is trained on the resulting pairs. If this interpretation is correct, it would be helpful to describe the training pipeline more explicitly (e.g., via pseudocode or an algorithm box), since the current presentation makes it difficult to understand how the different components interact.

- **Clarification of the training and evaluation protocol.** From the description, the dynamics model appears to be trained by maximizing the likelihood of one-step transitions using an Euler–Maruyama discretization, where the targets $(\overline z_{t+1}, \hat z_{t+1})$ are obtained by applying the encoder to the ground-truth microscopic states $X_{t+1}$. If this interpretation is correct, the model effectively learns transition dynamics between encoded states. However, it remains unclear how predictions are generated during evaluation. In particular, it would be helpful to clarify whether the encoder is used only to initialize the state $(\overline z_0,\hat z_0)$, after which the dynamics model generates trajectories autoregressively, or whether the encoder is also applied at later time steps using the ground-truth microscopic states. A clearer description of this procedure would improve the transparency of the experimental setup.

- The paper states that the autoencoder is trained only on the initial states $X_0$. If this is the case, the encoder may only observe a restricted subset of the state distribution during training. However, the encoder is later applied to point clouds from later time steps along the trajectories, whose distributions may differ substantially from the initial configurations. It would therefore be helpful to clarify whether the encoder is trained only on $X_0$ or on point clouds from all time steps, and whether the authors observed any issues related to distribution shift when applying the encoder to later states.

Overall, the paper addresses an interesting problem and introduces a technically appealing framework. However, the current presentation of the training and evaluation pipeline makes it difficult to fully assess the methodology and reproduce the approach.

---

> ### Author Rebuttal · Authors · 2026-03-31
>
> We thank the reviewer for the careful reading and practical suggestions. We address the concerns and questions below and will incorporate the reviewer’s suggestions to improve clarity in the revision.
>
> **On the training procedure.**
>
> Our current implementation follows a two-stage training procedure, as summarized by the reviewer. We agree that an algorithm box would improve clarity and will include it in the revision.
> Moreover, the proposed framework also allows the joint optimization. One can first train the autoencoder and the dynamics together, and then freeze the encoder and only fine-tune the dynamics. For example, we tried 100 epochs of joint training followed by 150 epochs to fine-tune the dynamics model. This achieves similar test performance to the two-stage training reported in the paper.
>
> **On the training and evaluation protocol.**
>
> Yes, the dynamics model is trained to maximize the likelihood of one-step transitions in our current implementation. During training, the encoder is applied to every time step to extract the $\hat{z}_t$ at $t=0, 1, ..., T$. During evaluation, the trained encoder is used only at the initial time to construct the state $\hat z_0$. After that, the learned dynamics model generates the trajectory *autoregressively*. Moreover, the framework also supports training on multi-step prediction, which in principle could improve the long-term prediction accuracy (e.g., [1]). We use the one-step transition for training because it is relatively easy to implement and already achieves good results.
>
> **On training the autoencoder only on $X_0$.**
>
> We apologize for this ambiguity. The autoencoder is trained on the full collection of microstates, not only on $X_0$. Thus, when the encoder is later applied to $X_t$ at later times to generate training data, it does not face the train-test mismatch caused by having seen only initial states. What is true is that, during evaluation, only the initial microscopic state  $X_0$ is used to compute the initial latent state.
>
> **Whether density reconstruction aligns with closure learning.**
>
> It is an important question. Our motivation is that the successful reconstruction of microstates from the latent variable provides a sufficient condition for the latent representation to act as the closure variable for modeling macro-dynamics. Prior closure-learning approaches often use pointwise reconstruction for exactly this reason (e.g., [2]). Our method follows the same principle, but replaces pointwise reconstruction by distributional reconstruction, which is naturally aligned with permutation invariance and avoids dependence on particle ordering.
>
> **On the scope of applicability.**
>
> This is also a good question. More precisely, our method applies when the available high-dimensional observation level can be modeled as evolving under approximately Markovian dynamics. Particle systems are a natural example of this setting, and we make the same assumption for the image data. Particularly, it can deal with the set-represented microstates without ordering, which is extremely challenging for previous closure modeling methods. In this sense, we believe our method can apply if the given aggregated or coarse-grained observations satisfy some inherent Markovian dynamics and we can reconstruct the observation information from a low-dimensional vector.
>
> Besides, our formulation applies more broadly than methods based on the reconstruction of the exact microscopic states. Such approaches require access to the exact state of each microscopic element (e.g., each particle). In contrast, our method reconstructs the \emph{distributional information} rather than the exact microstate itself. It therefore only requires the distribution induced by the observations, which is often more accessible in practice. For example, in imaging-based experiments, particle positions are typically inferred from blurred measurements induced by the imaging system’s point-spread function, rather than observed as exact positions [3].
>
> **On computational complexity and scaling.**
>
> Due to the word limit, we refer the reviewer to our response to the same question raised by Reviewer jjAj.
>
> **References:**
>
> [1] Rao, Pratyaksh Prabhav, et al. "Learning long-horizon predictions for quadrotor dynamics." 2024 IEEE/RSJ International Conference on Intelligent Robots and Systems (IROS). IEEE, 2024.
>
> [2] Bakarji, Joseph, et al. "Discovering governing equations from partial measurements with deep delay autoencoders." Proceedings of the Royal Society A: Mathematical, Physical and Engineering Sciences 479.2276 (2023).
>
> [3] Erdélyi, Miklós, et al. "Origin and compensation of imaging artefacts in localization-based super-resolution microscopy." Methods 88 (2015): 122-132.

---

> > ### Author Rebuttal · Reviewer_q6sW · 2026-04-02
> >
> > The rebuttal is clear and addresses several of my main concerns. In particular, the authors clarify the training pipeline, confirm that the encoder is trained on the full set of microstates, and provide a more precise description of the dynamics training and evaluation protocol. These clarifications improve the transparency of the method and resolve some of the ambiguities present in the original submission.
> >
> > However, some concerns remain only partially addressed. In particular, the connection between the density-reconstruction objective and the goal of learning effective closure variables is still not fully justified beyond analogy to prior work, and remains somewhat intuitive. Additionally, while the authors provide a breakdown of the computational cost, a more thorough discussion of scalability and optimization stability is still missing.
> >
> > Overall, the rebuttal strengthens the paper by resolving key issues related to clarity and methodology. While some conceptual and practical concerns remain, I update my assessment accordingly and raise my score from weak reject to weak accept.

---

> > > ### Author Response · Authors · 2026-04-07
> > >
> > > We thank the reviewer for the clarification. Let us address the remaining concerns more clearly below.
> > >
> > > **On the connection between the density-reconstruction objective and closure modeling**
> > >
> > > In general, closure modeling aims to summarize the high-dimensional microscopic information into a low-dimensional latent vector, which can be used as the state vector for modeling the system's macroscopic dynamics. Different methods mainly differ in the way of "summarizing microscopic information". Compared to prior closure modeling approaches, we use the density-based reconstruction as the surrogate for this purpose, in order to deal with microstates without ordering. In this sense, the role of the reconstruction is the same. *Only the representation of the microstate changes from an ordered list to a density.* The density-based representation allows us to learn latent variables at a controllable resolution, which can be better suited to modeling macroscopic dynamics. The reason is that, through the resolution parameter $\epsilon$, we can control at which level to retain microscopic information. Specifically, a larger $\epsilon$ smooths out very fine details in the reconstruction target where such fine details may be unnecessary for macroscopic dynamics modeling in a specific problem, while a smaller $\epsilon$ preserves more detailed information in the learned latent variable. In the limit $\epsilon \rightarrow 0$, the density-based representation reduces to the pointwise representation. Therefore, by varying $\epsilon$, density-based reconstruction can produce latent variables that encode microstates at different resolutions, and we select the one that is most suitable for modeling the macroscopic dynamics of a given system.
> > >
> > > To summarize, the underlying idea of closure modeling remains the same: summarizing microstates into the latent representation used for downstream macroscopic dynamics modeling. Our method differs from prior closure modeling in representing microstates by densities. When $\epsilon$ is very small (e.g., $\epsilon \rightarrow 0$), our density-based surrogate reduces to the pointwise information. When $\epsilon$ is large, it smooths out some microscopic details. And we can empirically choose the resolution that is most suitable for downstream macroscopic dynamics modeling in each task.
> > >
> > > **On scalability and optimization stability**
> > >
> > > 1. Scalability (how the proposed density-based decoder affects computational complexity). The computational complexity of evaluating our reconstruction objective is $\mathcal{O}(1)$ in the input size, i.e., it is independent of the number of particles. This is because the density-based reconstruction objective operates on Monte Carlo (MC) samples and generating these samples is also $\mathcal{O}(1)$ in the number of particles under our construction (note that all local kernels have the same weight and the same variance by our construction; so that we can firstly uniformly sample the mixture component and then sample from local Gaussian). This avoids the point-matching issue that often arises in permutation-invariant losses.
> > >
> > > 2. Optimization stability. The optimization stability also depends on the MC samples. Our density-based reconstruction objective (Eq.3) is estimated by MC samples, and we know the Monte Carlo estimator is unbiased. However, as in standard Monte Carlo objectives, using too few samples increases the variance of the gradient estimate and can make optimization noisier. In practice, this leads to the usual tradeoff between variance and computation: increasing the number of MC samples improves the optimization stability, but at additional computational cost. Therefore, when training is unstable, one can increase the number of MC samples. Or, in a simple way, we could use as many MC samples as are affordable, taking into account both the cost of evaluating the samples and GPU memory usage.
> > >
> > > We thank the reviewer for posting these helpful questions again and will include the discussion about scalability and optimization stability in the revision.

---

### Official Review · Reviewer_AsbL · 2026-03-15

**Soundness:** 3
**Presentation:** 4
**Significance:** 3
**Originality:** 3
**Overall Recommendation:** 4
**Confidence:** 4

**Summary:**

This paper studies the problem of inferring macroscopic dynamics from microscopic observations. To do this the authors propose a permutation invariant autoencoder pipeline, that works for inputs in an arbitrary ordering. The key innovation is to encode to an invariant latent representation, and compute the reconstruction loss over a distribution representation of the input and reconstructed output. The authors demonstrate the utility of their approach on several synthetic datasets showing performant macroscopic dynamics prediction.

**Compliance With Llm Reviewing Policy:**

Affirmed.

**Final Justification:**

Please see rebuttal acknowledgement

**Key Questions For Authors:**

“without a specified ordering, there is no natural mechanism to reconstruct the microstate with the index-wise loss.” why not use an equivariant architecture (transformers, graph neural networks) as the basis for the autoencoder?

**Limitations:**

yes

**Strengths And Weaknesses:**

**Soundness**. The submission appears technically sound. The authors clearly validate their claim that using an invariant representation, with a permutation insensitive loss outperforms architectures with a permutation sensitive encoding function.

**Presentation**. The submission is clearly written and well structured.

**Significance**. The approach is certainly interesting. I'm interested in comparisons to permutation equivariant baselines (generalization of invariance) such as autoencoding gnns, and transformers.

**Originality**. The work appears original.

---

> ### Author Rebuttal · Authors · 2026-03-31
>
> We thank the reviewer for the positive assessment and for this insightful question.
>
> We agree that permutation-equivariant architectures such as GNNs or transformers are important alternatives for learning on sets. Our goal in this work, however, is to learn a single low-dimensional permutation-invariant latent vector that summarizes the microscopic state for downstream macroscopic dynamics. In this setting, equivariant models do not by themselves resolve the main issue: while they process unordered inputs in a principled way, they typically produce particle-level outputs that still require an additional invariant aggregation step to obtain a compact closure representation. When used in an autoencoding pipeline, they are still paired with pointwise reconstruction, which reintroduces the difficulty of reconstructing an unordered set from the low-dimensional latent vector.
> By contrast, our method reconstructs the distribution induced by the point set, using a density-level objective that avoids pointwise alignment and naturally accommodates varying particle numbers. In this sense, the main contribution is not to advocate for a particular invariant encoder, but to formulate closure-variable learning through distribution reconstruction.
>
> More broadly, if the reviewer’s question is whether equivariant layers could be incorporated into our encoder, the answer is yes. One could replace the pointwise MLP blocks in DeepSets with a more expressive equivariant architecture, provided that the encoder ultimately outputs a low-dimensional permutation-invariant latent vector. This would be fully compatible with our framework. Our emphasis in the current paper is that the key novelty lies in the distribution-reconstruction formulation, rather than in the specific encoder family.
>
> We agree that this statement in the current paper can be made more precise. A better wording is: “without a specified ordering, there is no natural mechanism to reconstruct the microstate from a single low-dimensional representation using an index-wise loss.” We will revise it in the final version.

---

> > ### Author Rebuttal · Reviewer_AsbL · 2026-04-04
> >
> > Thank you for clarifying the motivation. My question about how the approach would compare with an equivariant architecture still stands. As you say, equivariant architectures can be used in settings where invariant representations are needed. Moreover, they are known to be more expressive, have better learning outcomes and require less training data than invariant architectures. I understand however, that these are not the focus of this work. My score remains positive.

---

> > > ### Author Response · Authors · 2026-04-07
> > >
> > > We thank the reviewer for the clarification. We now understand the point as suggesting an empirical comparison with stronger permutation-equivariant architectures, rather than only the justification for using equivariant or invariant architectures as the encoder.
> > >
> > > We recognize this is actually a good suggestion because it can provide clear evidence that the encoder in the proposed framework is not limited to DeepSet, but can be any encoder that ultimately produces a permutation-invariant latent variable. To test this, we replaced the DeepSets encoder in our current implementation with a Set Transformer [1], a representative permutation-equivalent architecture, for the interacting particle system. We use the original implementation provided by [1] with its default hyperparameter settings. All other settings are the same as our current implementation with DeepSet (e.g., the $\epsilon$, latent variable dimension, etc.).
> > > As in Sec. 4.2, we evaluate on three test cases:
> > >
> > > -- In-dst: same particle number and same initialization distribution as training;
> > >
> > > -- Diff-init:: same particle number but different initialization distribution;
> > >
> > > -- Diff-N: different particle number but same initialization distribution.
> > >
> > > The rollout prediction errors of the macroscopic dynamics are as follows (mean and standard deviation are from three runs):
> > >
> > >
> > >
> > > | Model | In-dst | Diff-init | Diff-N |
> > > |---|---:|---:|---:|
> > > | DeepSet | $(5.2 \pm 0.25) \times 10^{-5}$ | **$(4.4 \pm 0.19) \times 10^{-4}$** | $(5.2 \pm 0.46) \times 10^{-5}$ |
> > > | Set Transformer | **$(3.2 \pm 0.48) \times 10^{-5}$** | $(1.7 \pm 0.24) \times 10^{-3}$ | **$(2.6 \pm 0.23) \times 10^{-5}$** |
> > >
> > >
> > > The Set Transformer improves over DeepSets on the In-dst and Diff-N settings, while DeepSets performs better under distribution shift. We think that such results are exactly because Set Transformer is more expressive, which improves performance when test configurations remain close to the training distribution, but may also make the learned latent representation more specialized to the training distribution. Therefore, the Set Transformer encoder performs better when the testing data distribution is similar to training (i.e., In-dst and Diff-N), but can be less robust when the initialization distribution itself changes (i.e., Diff-init).
> > > Besides, these results empirically demonstrate that the paper's main contribution is orthogonal to the encoder choice, i.e., our framework is compatible with stronger set encoders.
> > >
> > > Moreover, it would also be valuable to compare DeepSet and Set Transformer with different input sizes. As pointed out by the reviewer, the equivariant architecture usually requires less training data, and data efficiency is an important consideration in real applications. However, due to the time limit of this rebuttal and computing resources, we are not able to finish that experiment. We would like to include it as well in the final revision.
> > >
> > > **Reference:**
> > >
> > > [1] Lee, Juho, et al. "Set transformer: A framework for attention-based permutation-invariant neural networks." International conference on machine learning. PMLR, 2019.

---

### Decision · Program_Chairs · 2026-04-30

**Decision:**

Accept (regular)

**Comment:**

This paper introduces a permutation-invariant autoencoder framework designed to infer macroscopic dynamics from microscopic particle systems without requiring a canonical ordering of microstates. The key technical innovation is a distribution-reconstruction objective that represents point sets as kernel density estimates, allowing the model to naturally handle varying particle numbers and avoid expensive point-matching losses like Chamfer distance. Reviewers lauded the method's theoretical soundness, clear presentation, and empirical effectiveness across diverse settings including interacting particle systems and polymer dynamics.

The authors provided a comprehensive rebuttal that addressed all reviewer questions. They performed a systematic numerical sensitivity analysis for the smoothing scale and latent dimension. Furthermore, they clarified the training protocol and demonstrated the framework's compatibility with more expressive set encoders like the Set Transformer. Consequently, the reviewers reached a positive consensus, noting that the clarifications resolved important ambiguities and significantly strengthened the submission.

Therefore, the final recommendation is to accept the paper.